



# The nodal dependence of long-period ocean tides in the Drake Passage

Philip L. Woodworth[1], Angela Hibbert[1]

[1]National Oceanography Centre, Joseph Proudman Building, 6 Brownlow Street, Liverpool L3 5DA, United Kingdom

*Correspondence to*: Philip L. Woodworth (plw@noc.ac.uk)

**Abstract.** Almost three decades of bottom pressure recorder (BPR) measurements at the Drake Passage, and 31 years of hourly tide gauge data from Vernadsky station on the Antarctic Peninsula, have been used to investigate the temporal and spatial variations in this region of the three main long-period tides Mf, Mm and Mt (in order of decreasing amplitude, with periods of a fortnight, a month and third of a month respectively). The amplitudes of Mf and Mt, and the phase lags for all three constituents, vary over the nodal cycle (18.61 years) in essentially the same way as in the equilibrium tide, so confirming the validity of Doodson's 'nodal factors' for these constituents. The amplitude of Mm is found to be essentially constant, and so inconsistent at the three-sigma level from the ±13% (or ~±0.15 mbar) anticipated variation over the nodal cycle, which can probably be explained by energetic non-tidal variability in the records at monthly timescales and longer. The north-south differences in amplitude for all three constituents are consistent with those in a modern ocean tide model (FES2014), as are those in phase lag for Mf and Mt, while the difference for Mm is smaller than in the model. BPR measurements are shown to be superior to conventional tide gauge data in such tidal studies, thanks to the lower proportion of non-tidal variability in the records.

## 1 Introduction

The ocean tide at each location is usually represented as a combination of harmonic constituents with frequencies corresponding to those of lines in the tidal potential (Cartwright and Tayler, 1971; Cartwright and Edden, 1973). The major lunar constituents are always accompanied by sidebands separated in frequency by $\pm$ 1/18.61 cycles per year, 18.61 years being the nodal (or draconic) period of regression in the mean longitude of the lunar ascending node (Doodson and Warburg, 1941). The most efficient way of accounting for the sidebands in a harmonic expansion is via the use of 'nodal factors' $f$ and $u$, whereby the simple representation of a single constituent:

$$Hcos(\omega t + A - G)$$

[1]

in which $\omega$ is the angular frequency of the constituent, $H$ and $G$ are its amplitude and phase lag, and $A$ is its astronomical argument at time $t = 0$, is modified to:



$$fHcos(\omega t + A + u - G)$$

[2]

where $f$ and $u$ are time-dependent functions of the longitude of the ascending node ($N$). For example, in the tidal potential
(or equilibrium tide), the main lunar semidiurnal tide (M2) has nodal factors:

$$f = 1.0 - 0.037\,cos(N)\,, u = -2.1°\,sin(N)$$

[3]

retaining only terms in $cos(N)$ and $sin(N)$, and neglecting smaller terms depending on $cos(2N)$ etc. (Doodson, 1928;
Doodson and Warburg, 1941; Pugh and Woodworth, 2014).

Because the frequencies of the sidebands are similar to the constituent's central frequency, it is usually assumed that
the response of the ocean at the sidebands and at the central frequency will be in proportion to that given in the tidal potential
i.e. that the same admittance will apply. However, nodal factors different to expectations from the tidal potential (or
equilibrium tide) have been found at many locations, at least for semidiurnal tides.

For example, smaller values of $f$ for M2 were found around the UK by Amin (1983, 1985) and were explained as
being a consequence of non-linear frictional damping. Similar findings were obtained from measurements of mean tidal
range around the UK by Woodworth et al. (1991). Differences from the expected nodal factors were found in data from the
west coast of Australia (Amin, 1993) and the Bay of Fundy and Gulf of Maine (Ku et al., 1985; Ray, 2006; Müller, 2011).
Feng et al. (2015) found differences for both semidiurnal and diurnal tides at locations along the coast of China. In a survey
of long-term changes in the amplitudes and phase lags of the four main tidal constituents around the world (M2, S2, O1 and
K1), Woodworth (2010) pointed to many locations where differences in $f$ from those expected from the equilibrium tide
were evident.

Turning to the long-period tides, all of the long-period constituents of the equilibrium tide have amplitudes
proportional to $\left(\frac{1}{3} - sin^2(latitude)\right)$ with no zonal dependence. The amplitudes are twice as large at the poles as at the
equator; they are 180° out-of-phase between high and low latitudes; and they have zero amplitude at 35° N/S. Proudman
(1960) suggested that, at least for the longest of the long-period tides (the 18.61 year nodal tide), the tide in the real ocean
should be a close approximation of its equilibrium form, and that still seems to a good theory (Woodworth, 2012). However,
tidal modelling and observations by tide gauges and satellite altimetry have demonstrated that the long-period tides with
shorter periods in the real ocean, such as Mf and Mm with periods of approximately a fortnight and a month respectively,
have significant spatial variations from their equilibrium form (Wunsch et al., 1997; Mathers and Woodworth, 2001; Egbert
and Ray, 2003; Ray and Egbert, 2012; Ray and Erofeeva, 2014).

Although the spatial variations of the long-period tides are now much better understood, it is also of interest to
consider whether their temporal (nodal) variability conforms to expectations. The Mf constituent (period 13.66 days) is



particularly interesting in this regard. In the equilibrium tide, Mf is the largest of the long-period tides and has very large nodal variations:

$$f = 1.043 + 0.414\,cos(N), u = -23.7°\,sin(N)$$

[4]

Why the first term in $f$ is not identically 1.0 (for Mf and for many other constituents) arises from the way that Doodson (1928) combined sideband constituents in order to provide simple functions in terms of $N$ only. Doodson's nodal parameterisations, especially those for Mf, are discussed in the Appendix.

As far as we know, the magnitude of this temporal variability for the long-period tides in the real ocean has never
been verified properly. In principle, one would have expected that the relatively large amplitude and short period of Mf would have enabled the temporal variation of its amplitude and phase lag to be estimated reliably from two decades of tide gauge data. However, there is always non-tidal background variability at fortnightly timescales to contend with. Most research on long-period tides in tide gauge records has been focused on regions such as the low-latitude Pacific, where the non-tidal background is much less than at higher latitudes (e.g. Miller et al., 1993). However, the long-period tides are also
small in these regions (i.e. centimetric, see Figure 5 of Ray and Egbert, 2012). These studies of Pacific data were primarily concerned with establishing how the non-equilibrium aspects of Mf and Mm varied spatially, rather than temporally (Wunsch, 1967). Even though some long tide gauge records exist at high latitudes (e.g. northern Norway or Canada), where long-period amplitudes are larger, the relatively high background of non-tidal sea level variability means that it is difficult to make an accurate determination of the long-period tides without also modelling the non-tidal background (e.g. Crawford,
1982).

In this paper, we report on the temporal variations of the amplitudes and phase lags of Mf, Mm and Mt (period of one third of a month) at the Drake Passage to see if they are consistent with equilibrium expectations. These are the three long-period tides, in order of decreasing amplitude in the equilibrium tide, that one is likely to be able to extract from records of about one year. The Drake Passage is at a sufficiently high latitude that any long-period tides should be larger than in
most parts of the ocean. In addition, our investigation is based on the use of measurements of bottom pressure (BP) obtained over almost three decades, instead of on conventional coastal tide gauge data. It will be seen that bottom pressure recorders (BPRs) are inherently more suitable for providing long-period tidal information than coastal tide gauges. However, as a comparison of different measurement techniques, we also make use of 31 years of hourly sea level data from the nearby Vernadsky station, which has the longest tide gauge record in Antarctica.



## 2 Bottom pressure recorder data and methods

Cartwright (1999, chapter 13) provides a history of the development of BPRs, primarily by groups in Germany, France, USA and UK. Cartwright himself and colleagues from the National Oceanography Centre (NOC, as it is now called) made extensive use of BPRs in sets of 'pelagic' (pertaining to the open sea) tidal measurements, first in waters around the UK, and then throughout the Atlantic Ocean (Cartwright et al., 1988; Spencer and Vassie, 1997). The same equipment was also used in international studies of non-tidal ocean processes (e.g. Cartwright et al., 1987), culminating in the late 1980s in the deployment of BPRs at Drake Passage in order to monitor fluctuations in the transport of the Antarctic Circumpolar Current (ACC) as part of the World Ocean Circulation Experiment (Woodworth et al., 1993).

Most of the BPR deployments were made on the north and south sides of Drake Passage in order to measure changes in the pressure gradient between them. Bottom landers based on the 'Mk.IV' or similar designs were used in most cases (Figure 1a, Spencer and Vassie, 1997). Over half of the deployments took the form of recoveries and redeployments on an annual basis at depths around 1000 metres, providing records of 15-minute average bottom pressure typically one year long. The other half of the deployments were made at greater depths, between 2000-4000 metres. Three deployments were made using the longer duration Multi Year Return Time Level Equipment (MYRTLE) instrument that provided BP records approximately 4 years long (Figure 1b). The measurement programme was terminated in 2016, resulting in a BP data set spanning almost three decades.

The measurements have been used in studies of ACC variability, as reviewed by Meredith et al. (2011). BP measured at the south side of the Drake Passage has been shown to be particularly useful as a monitor of fluctuations in ACC transport (Hughes et al., 2003; Hibbert et al., 2010). The data have even proved to be useful in studies of tsunami travel times (Rabinovich et al., 2011). As regards tides, the data have been employed in validation studies of models of the semidiurnal and diurnal tides observed by satellite altimetry (Ray, 2013).

BP has advantages in tidal studies over sea level recorded by conventional tide gauges at the coast. An obvious factor is that BPRs can be deployed in deep water off-shore (pelagic), at some distance from where storm surges and other shallow-water processes are largest. Another factor is that much of the sea level variability due to air pressure changes is compensated automatically by air pressure itself in the bottom pressure measurement (the inverse barometer effect). As a result of these two factors, BP records tend to have a smaller percentage of non-tidal variability (or 'noise') than do tide gauge records.

The main disadvantages of a BP record are instrumental drift (also known as 'creep') and the absence of a geodetic datum. Fortunately for tidal studies, creep is primarily a low-frequency process that tends not to impact upon the determination of high-frequency components of the record, such as the semidiurnal and diurnal tides (Watts and Kontoyiannis, 1990; Spencer and Vassie, 1997). Instrumental drift does tend to preclude the reliable observation of annual and semiannual tides in BPR data. However, these long-period tides are not of lunar origin and so are not the concern of the present investigation. The absence of a datum is an important factor when it is required to combine individual yearly records into longer, continuous records. Unless overlapping records are available, from which the datum of one deployment can be



related to that of another, then techniques such as 'end point matching' have to be employed (e.g. Meredith et al., 2004). Although we make use of one such combined record below, in order to demonstrate clearly the existence of long-period tides in the data, combinations of records are not required for most of the present study in which we analyse the records from each deployment separately.

Almost all the Drake Passage BPR data obtained by NOC since 1992 have been re-analysed recently as part of a Natural Environment Research Council (NERC) project called 'Weighing the Ocean'. (Several NOC deployments in the centre of the Passage and in the Scotia Sea were not included.) Data were subjected to a new set of quality control that identified any suspect measurements and corrected as far as possible for timing uncertainties and instrumental drift. The processed data, consisting of records from 35 individual deployments, are available on the web site of the Permanent Service

for Mean Sea Level (PSMSL) (http://www.psmsl.org). Ten other records were added from deployments before 1992 at Drake Passage and from the Falkland-Signy (F-S) line (Woodworth et al., 1996). These earlier records can be obtained from http://www.ntslf.org/files/acclaimdata/bprs/.

      Figure 2 shows the locations of the 45 deployments, many of which were at essentially the same positions and so overlap on the map. The 35 locations with reanalysed data from the PSMSL web site include those on the north and south

sides of the Passage south of the Falkland Islands, and that of the first of the three MYRTLE deployments close to Signy Island. The two other MYRTLE deployments were also made on the south side, but in more central positions. The ten earlier deployments include the western-most north-south pair and those on the F-S line to the east.

      We have treated the data from each deployment as a separate record, with record lengths from 296 to 1470 days. Each record of BP was first subjected to a tidal analysis consisting of typically 57 semidiurnal, diurnal and higher-frequency

constituents, the exact number of constituents depending on the record length. However, importantly, long period tides were not included in the tidal analysis. Residuals of the analysis were interpolated to hourly values, and simple arithmetic averaging of the 24 hourly residuals each day provided the time series of daily mean values of BP that are discussed below. For full details of the data processing, see http://www.psmsl.org/data/bottom_pressure/processing_procedures.php.

      The evidence for long-period tides in the BP data is demonstrated clearly in Figure 3(a). In this case, the individual

time series of daily mean BP were de-meaned and de-trended and, when daily values were available from more than one location on the same day (i.e. from both north and south sides of the Passage), they were averaged, so providing a continuous, composite time series spanning over 26 years. Figure 3(a) shows the resulting power spectrum which indicates clearly the presence of Mm (period 27.55 days), MSf (14.77 days), Mf (13.66 days) and Mt (9.13 days). Such tidal signals are obviously less well resolved when analysing records individually (Figure 3(b)). In this case, we are dealing with records

of different lengths, at times when the relative proportions of each long-period component (primarily Mf) will be different, and when there will be different proportions of tidal and non-tidal variability. Consequently, spectra were produced for each individual record, normalised to have unit energy in the long-period tidal band (0.02-0.15 cpd), and then averaged into bins of 0.005 cpd, so providing a spectrum 'typical' of an individual record. It can be seen that Mf, and to a lesser extent Mm, are still present, while Mt is less well resolved, and MSf cannot be seen above the background.



MSf is an interesting constituent that occurs for two reasons. It is partly a long-period tide in its own right, with an amplitude in the equilibrium tide 8.7% that of Mf, and with variations in amplitude through the nodal cycle of ±14%. It is also partly an interaction constituent (see below), with a nodal variation of ±3.7% as for M2 in Equation 3. However, its generally low amplitude suggests that verification of its nodal variation in real data will be much harder than for Mm and Mt, and we have not considered MSf in detail further.

In order to study the time-dependence of the long-period tides, their amplitudes and phase lags were determined for each deployment record independently, by means of a regression of the daily means of BP in terms of three harmonics with periods of Mf, Mm and Mt plus a linear trend. The three periods are so different that the amplitudes and phase lags determined for each harmonic are almost the same whether the regression includes all three constituents or each one individually. This procedure assumes that the amplitudes and phase lags of each harmonic do not change during the record i.e.

$$h(t) = H_i cos(\omega t - \varphi_i)$$

[5]

where $h(t)$ is BP for a particular harmonic that is a function of time $t$ measured from the start of 1988, $\omega = \left(\frac{2\pi}{period}\right)$ radians per day, and $H_i$ and $\varphi_i$ are the amplitude and phase lag from the regression for deployment $i$. Therefore, one can investigate how $H_i$ varies as a function of the central date of each record ($T_i$), and similarly from Equations 2 and 5 one can relate:

$$\varphi_i + A = G - u_i$$

[6]

where the variation of $\varphi_i$ as a function of $T_i$ can be described by an oscillation ($u_i$) around the average phase lag ($G$). $A$ is the astronomical argument for the harmonic constituent concerned at the start of 1988. If the start of that year is defined by GMT (UT), then $G$ will be the constituent's Greenwich phase lag.

The regression is made using the G02CGF function of the Numerical Algorithms Group (NAG) library (https://www.nag.co.uk). This results in the determined amplitudes for the three harmonics having the same standard errors, while standard errors on each phase lag are defined by the standard error on the amplitude divided by the amplitude itself (times $360°/2\pi$). The same standard error for each amplitude arises from an assumption of white noise in the residuals of the regression. Consequently, they may be potentially estimated too low (see below). However, inspection of the scatter in the plots below, compared to the formal errors, demonstrates that they will have been estimated fairly reliably.



## 3 Results for Mf, Mm and Mt

### 3.1 BPR data

In this section, we discuss findings for Mf, Mm and Mt obtained from the BPR data. Figure 4(a) shows an example of one of

the records of daily mean BP and the result of a regression fit in terms of the three harmonics. In fact, this is a particularly good example of a record from the north side of the Drake Passage with a relatively small proportion of non-tidal variability, at a time (2008-9) when the amplitude of Mf was larger than average. It serves to make the point that information on the amplitude and phase lag of Mf can be extracted reliably from such records.

       Figure 2 shows that the deployments in the Drake Passage took place over a large area. However, we can take

advantage of the fact that the spatial scale of variation in Mf, and of other long-period tides, is also large (e.g. see Figure 5 of Ray and Egbert, 2012 and the discussion of the FES2014 model in Section 4). Consequently, as a first approximation, all of the values of $H_i$ and $G_i$ from the many deployments can be considered as having been obtained at the same location.

       Figure 5(a,b) presents the amplitude and phase lag of Mf respectively, obtained from the harmonic analysis of each record. The amplitude units are mbar which can be taken as being approximately equivalent to cm of seawater. A clear nodal

(18.61 year) variation can be seen in the amplitudes (Figure 5a), with the red line showing a fit in terms of $\cos(N)$, constrained to peak when $N = 0$ at 2006.5. The red line passes equally well through the black and blue points, representing deployments on the north and south sides of Drake Passage respectively

       The mean amplitude in the fit is $2.18 \pm 0.04$ mbar, and the amplitude of the nodal variation is $0.93 \pm 0.06$ mbar, or $43 \pm 3$ % of the mean value. This may be compared to the 40% expected from the equilibrium tide (i.e. 0.414/1.043 in

Equation 4).

       If the real Mf had a spatial variation similar to its counterpart in the equilibrium tide, one could adjust the measured amplitudes for the difference in latitude of the various deployments (an equilibrium long-period tide has no variation zonally). Consequently, if the amplitudes in Figure 5(a) are multiplied by:

$$\left(\frac{1}{3} - sin^2(reference\ latitude)\right) / \left(\frac{1}{3} - sin^2(latitude)\right)$$

[7]

where a reference latitude of 58 °S is chosen in the middle of the Drake Passage, then one obtains Supplementary Figure 1. There is a larger scatter about the fit than in Figure 5(a), with a chi-square three times as large. Most of the north-side values (black) are now systematically larger than the south-side values (blue), a result which is inconsistent with Mf amplitudes

having the same latitude dependence as in the equilibrium tide.

       Figure 5(b) shows the variation in phase lag obtained from each record i.e. the variation in values of $(\varphi_i + A)$ or $(G - u_i)$. The red line shows a fit in terms of $sin(N)$, with the nodal variation constrained to be 0° when $N = 0$. The mean value in the fit is $191.9 \pm 1.0°$ while the amplitude of the sinusoidal variation is $28.4 \pm 1.4°$, which is a little larger than



equilibrium tide expectations (Equation 4). The black and blue points are clearly separated, indicating a phase lag on the south side of the Drake Passage $22 \pm 2°$ larger than on the north side (obtained by weighting the individual observed phase lags minus the fitted phase lag by the reciprocal of the square of the standard error on the phase lag). Once again, this is inconsistent with the equilibrium tide, in which both sets would have a phase lag of 180° at these latitudes.

The next largest long-period tide one can investigate is Mm. This represents more of a challenge, with a longer period (27.55 days) and an amplitude in the equilibrium tide that is approximately half that of Mf. In addition, it has a nodal variation in its equilibrium amplitude that is about a third that of Mf:

$$f = 1.0 - 0.130\,cos(N), u = 0.0$$

[8]

Figure 4(b) shows an example of a BP record from the south side of the Drake Passage, at a time (1999-2000) when the amplitude of Mf was much less than in Figure 4(a), indicating that Mm can be readily identified by eye at such times. Therefore, we can have some confidence in the harmonic fitting.

      Figure 6(a) shows the observed variation in Mm amplitude with no obvious differences between north and south
side values. Once again, the red line shows a cosine fit to the amplitude values. The mean amplitude is $1.34 \pm 0.04$ mbar. However, the amplitude of the cosine is close to zero at $0.00 \pm 0.06$ mbar, or $0.1 \pm 4.2$ % of the mean value (with the correct negative sign of Equation 8). This is much less than the 13% expected from the equilibrium tide, so there is approximately a 3-sigma difference between measurements and expectations.

      The individual phase lags obtained for Mm (Figure 6b) are similar on each side of the Drake Passage. However,
they have large uncertainties. Weighting each phase lag as for Mf above gives a south-north difference of $2 \pm 3°$. They have no evident nodal variation, as suggested by Equation 8. Therefore, in this case, instead of a nodal fit the red line in Figure 6(b) indicates the median phase lag of $177.3 \pm 4.4°$. This value is consistent with equilibrium expectations for a long-period tide at this latitude.

      The third long-period tide to be investigated is Mt (period 9.13 days). This is the next largest long-period tide in the
equilibrium tide, with an amplitude about one third that of Mm and one sixth that of Mf, and with a nodal variation in $f$ and $u$ similar to that for Mf in Equation 4. In this case, the amplitudes are so small that the contribution of Mt to the BP time series is not readily apparent by eye, such as in Figure 4(a,b), although Mt is undoubtedly present as shown by Figure 3(a,b). Therefore, in this case, one has to rely on the formal uncertainties provided by the regression fits.

      Figure 7(a) shows the amplitudes obtained for Mt, which are similar on the north and south sides of the Drake
Passage, with a mean value of $0.43 \pm 0.04$ mbar. The red line indicates a nodal variation with an amplitude of $0.12 \pm 0.06$ mbar, or $28 \pm 13$ % of the mean value, which is consistent with $f$ in Equation 4 within the uncertainties. Figure 7(b) shows the estimated phase lags from the analysis of each record. Phase lags have smaller uncertainties after 2001, which follows from the larger average amplitudes in the second half of the data (Figure 7a). They have an average value of $197.3 \pm 5.0°$. A



weighted fit indicates phase lags 22 ± 9° larger on the south side. A sinusoidal fit to all of the phase lag values considered together results in an amplitude of 30 ± 7°, consistent with Equation 4.

**3.2 Vernadsky data**

5 Vernadsky on the west coast of the Antarctic Peninsula (Figure 2) has the longest tide gauge record in Antarctica. The base is now operated by the National Antarctic Scientific Center of Ukraine. A float gauge was installed at the base (then called Faraday) at around the time of the International Geophysical Year (1957-58). Monthly mean sea levels are available from the PSMSL starting in 1958, while hourly values from March 1984 to December 2014 can be obtained from the Global Extreme Sea Level Analysis (GESLA) data set (http://www.gesla.org, Woodworth et al., 2017).

10 Vernadsky tide gauge data have been used in several studies of ACC variability alongside the information from the Drake Passage BPRs (Hughes et al., 2003; Woodworth et al., 2006). For present purposes, Vernadsky data enable an interesting comparison to be made on how much better Mf can be observed in BP measurements than in coastal tide gauge data.

Each year of hourly data from Vernadsky has been analysed in a similar way as described for the BP measurements, 15 providing daily values of sea level from which estimates of Mf amplitude and phase lag have been obtained. (We considered Mm and Mt to be below noise level in these one-year records.) Figure 8(a) shows the amplitude values, which have individual uncertainties approximately five times larger than for the BPRs in Figure 5(a). The mean amplitude in the plot is 2.90 ± 0.25 cm (and so the Mf harmonic constant would have an amplitude of 2.90/1.043 = 2.78 cm). This is larger than for the nearby BPRs. The nodal cycle shown in red has an amplitude of 1.20 ± 0.36 cm, or 41 ± 12 % of the mean value, almost 20 exactly the same as for the BPRs and again consistent with expectations from Equation 4. Phase lag (Figure 8(b)) is also consistent with the BP data, in having an average of 184.9 ± 4.7°. Within the large scatter from year to year, a nodal variation with an amplitude of 22 ± 7° can be just about discerned. (Five years of data with phase lags outside the plot limits were not used in this nodal fit.)

Comparisons of Figures 5 and 8 underline the point we wish to make regarding the superiority of BP measurements 25 to long-period tidal studies. It is possible that modelling of the non-tidal variability in the Vernadsky records in terms of a response to winds and air pressures could result in a more clearly identified Mf, but one doubts if it could ever be equally as good. Crawford (1982) provides an example of an attempt at such modelling in Canadian tide gauge data.

**4 Discussion**

30 Some of the findings of the previous section are consistent with expectations from the equilibrium tide, while those that are not require explanation.

As mentioned above, the long-period tides in the equilibrium tide have simple spatial distributions in amplitude and phase, with north-south variations only. However, their spatial distributions in the real ocean are now known to depart considerably from equilibrium expectations, with larger departures at shorter period (e.g. see Figure 2 of Ray and Erofeeva,



2014). These differences are most evident when contrasting the Pacific, Atlantic and Indian Ocean low- and mid-latitude basins.

If one considers Mf in particular, atlases of this constituent have been available for many years, notably since the data assimilation numerical modelling of Schwiderski (1982). More recent co-tidal distributions for Mf have been obtained

from altimeter measurements and models by Kantha et al. (1998, Figure 7), Mathers and Woodworth (2001, Plate 4) and Egbert and Ray (2003, Figure 1). These are consistent with Mf phase lag increasing when travelling south down the Pacific coast of South America, with the 180° contour around the Drake Passage, and with a complicated amphidromic pattern in the South Atlantic to the NE of the Falklands. More recent studies have included the development of the FES2004 ocean tide model, which also showed these features (Lyard et al., 2006, Figure 2), with roughly the same Mf amplitude on both sides of

the Drake Passage and larger phase lag on the south side than north side.

FES2014 (Finite Element Solution 2014) is the latest in the series of state-of-the-art global ocean tide models provided by French groups.  It provides elevations and currents (amplitude and phase) and tidal loading information for 34 tidal constituents on a global 1/16°x1/16° grid. FES2014 (2018) provides more detailed information.

Supplementary Figure 2(a,b) shows the Mf amplitude and phase lag for Mf at the Drake Passage from the FES2014

model. Some points of consistency with our findings are as follows. First, the model has much the same amplitude over the whole area (~2 cm), and phase lags are essentially zonal, largely justifying our decision to combine amplitudes and phase lags from all deployments in Figure 5, and the subsequent discussion in terms of north- and south-side values.

Second, we found the amplitudes for Mf to be similar on the north and south sides of Drake Passage (Figure 5a), but phase lags were shown to be 22 ± 2° larger for the southern deployments (Figure 5b). The latter is qualitatively consistent

with Supplementary Figure 2(b). Third, the 192° average phase lag for Mf from all the BPRs taken together (Figure 5b) is consistent with the ~190° contour in mid-Passage in Supplementary Figure 2(b). On the other hand, the 185° phase lag at Vernadsky is a little lower than the ~200° one would infer from Supplementary Figure 2(b).

If a tide model such as FES2014 was perfect, then any differences in observed amplitudes and phase lags due to the different deployment locations could be removed by relating each set of findings to those which would have been obtained at

a reference location, using an admittance relationship:

$$H_i^{ref} = H_i \frac{HM^{ref}}{HM_i}$$

[9]

where $H_i$ is the measured amplitude for deployment $i$, $HM_i$ and $HM^{ref}$ are the model amplitudes at the deployment and reference point locations respectively, and $H_i^{ref}$ is the inferred amplitude at the reference point. Similarly,

$$\varphi_i^{ref} = \varphi_i + \varphi M^{ref} - \varphi M_i$$





where $\varphi_i$ is the measured phase lag for deployment $i$, $\varphi M_i$ and $\varphi M^{ref}$ are the model phase lags at the deployment and reference point locations respectively, and $\varphi_i^{ref}$ is the inferred phase lag at the reference point. If the model represented the spatial dependence of the tide correctly, then $H_i^{ref}$ and $\varphi_i^{ref}$ should have only a temporal dependence.

Figure 5(c) shows the resulting model-adjusted values of Mf amplitude, using a reference point location of 57° W, 58° S, demonstrating satisfactory consistency between values north and south. That was already the case in Figure 5(a), and the similarity of Figures 5(a) and (c) reflects the uniformity of amplitude in the model in this area. The nodal fit in red shows a cosine with an amplitude of 43 ± 3 % of the mean, which is identical to that in Figure 5(a). For phase lag, Figure 5(d) demonstrates a considerable improvement on Figure 5(b), with values north and south in agreement (weighted south-north difference of 0 ± 2°). In addition, the nodal fit in red has an amplitude of 23.4 ± 1.4°, which is closer to Equation 4 than for Figure 5(b).

Consequently, the temporal variation of Mf can be seen from Figure 5 to conform closely to expectations from its equilibrium form shown by Equation 4. Mf has the largest amplitude of the long-period tides we have investigated, which together with its relatively short period compared to the typically 1-year long records, means that it is the best resolved. Our finding of consistency with equilibrium expectations parallels an observation regarding fortnightly variations in the solid earth in a study of polar motion data by Ray and Egbert (2012), who concluded that a similar admittance applied to Mf and its nodal sideband (see also earlier work by Gross, 2009). As explained above, the same admittance for a central frequency and its sidebands indicates that the nodal factors of the equilibrium tide apply equally as well to the tide in the real ocean (or solid earth).

Turning to Mm, its spatial variation in FES2014 is shown in Supplementary Figure 2(c,d). Once again, amplitudes are much the same over the whole area, and phase lag contours are roughly zonal. However, in this case, Supplementary Figure 2(d) indicates a north-south gradient of phase lag about half that for Mf in Supplementary Figure 2(b). Our observation of a small south-north difference of 2 ± 3° is qualitatively consistent with the smaller gradient in the model (a south-north difference of ~10°). The observed average phase lag of 177° for Mm from all deployments combined is a little lower than the ~185° contour in mid-Passage in Supplementary Figure 2(d).

Figure 6(a) shows that Mm amplitudes for the first decade are lower in the south, but they become more equal to the northern ones thereafter. One may note that five of the six deployments with particularly low amplitudes before 1994 are from the F-S line. However, some kind of general amplitude bias in these early deployments is unlikely, given that their corresponding amplitudes for Mf are consistent with later ones (Figure 5a). Overall, Figure 6(a) does not provide evidence for a temporal dependence of Mm amplitude similar to that of Equation 8. However, identifying a nodal signal of only ~0.15 mbar is clearly a challenge given the uncertainties. At least, the absence of any evidence for nodal variation in Mm phase lag (Figure 6b) is consistent with Equation 8.





A nodal variation in amplitude of ~0.15 mbar might be technically within the resolution of the BPR measurements if Mm was accompanied by only a limited amount of non-tidal variability on similar (monthly) timescales. Monthly timescales are more comparable to processes associated with ACC variability. Sheen et al. (2014) showed that eddy kinetic energy is more intense in the north of Drake Passage, where the main fronts and their meanders occur. However, eddy activity also occurs in the south. In addition, variability in BP in this region has a contribution on 30-70 day timescales from the Madden-Julian Oscillation (Matthews and Meredith, 2004), which could potentially impact on our determination of Mm.

An attempt was made to reduce the amount of non-tidal variability in the records with the use of 5-day values of BP from the Nucleus for European Modelling of the Ocean (NEMO) 1/12° ocean circulation model for 1988-2012 (Hughes et al., 2018), with the aim of better resolving any nodal tidal signals, particularly that for Mm. The model BP was found to have a high correlation with measured non-tidal BP for most of the southern deployments, while correlations were weaker in the north, as Sheen et al. (2014) would suggest. However, subtraction of the model values from the measurements resulted in little change in the determined Mm amplitudes and phase lags.

FES2014 model adjustments for Mm from Equations 9 and 10 result in Figure 6(c,d). Figure 6(c) confirms similar amplitudes north and south, and the nodal fit gives an amplitude of 0.8 ± 4.3% of the mean value, a little larger than that from Figure 6(a), but still 3-sigma away from expected in Equation 8. As for phase lag (Figure 6d), the weighted south-north difference is now -11 ± 3°, as can be readily observed by eye. This indicates that the model over-corrects for spatial variation in phase lag. This suggests that the difference in Mm phase lag across the real Drake Passage is less than in the model.

One might have expected the detection of Mt to be easier than that for Mm, thanks to its shorter period, even though it has a much smaller amplitude. Figure 7(a) shows an average amplitude of 0.43 mbar, with little evidence for differences between values north and south, while Figure 7(b) indicates an average phase lag of ~197°, and some evidence for phase lags about 22° larger in the south than in the north. The temporal variations in Mt amplitude and phase lag in Figure 7(a,b) are consistent with equilibrium expectations within the large uncertainties for this small constituent.

Supplementary Figure 2(e,f) gives the corresponding information for Mt from the FES2014 model. (This constituent is called Mtm in the model.) Supplementary Figure 2(e) shows an amplitude of ~0.4 cm over most of the area, while Supplementary Figure 2(f) shows a meridional gradient for phase lag similar to that obtained from the BPRs. The observed mean phase lag of 197° is consistent with the mid-Passage contour in Supplementary Figure 2(f).

If one applies the FES2014 model adjustments from Equations 9 and 10 to the observed amplitudes and phase lags for Mt, then one obtains Figure 7(c,d). This procedure results in apparent improvements as for Mf. Figure 7(c) is much the same as Figure 7(a), with similar amplitudes north and south. The nodal fit in Figure 7(c) has an amplitude of 30.5 ± 14.0 % which is similar to that obtained above for Figure 7(a). For phase lag, Figure 7(d) demonstrates an improvement on Figure 7(b) with a weighted south-north difference of -5 ± 9°, consistent with zero difference. The nodal fit shows an amplitude of 23 ± 7°, closer to Equation 4 than for Figure 7(b).

As an aside, one can mention that Mt is to some extent a 'forgotten constituent'. It is represented in harmonic expansions of the tidal potential (Doodson, 1921; Cartwright and Tayler, 1971) as a line with Doodson number 0,3,0,-1,0,0



(or 085.455 in Doodson's notation) with one major nodal sideband (0,3,0,-1,1,0). However, Doodson did not usually refer to it explicitly in his own papers (e.g. Doodson, 1928), and it is not included in the standard sets of harmonic constituents used in tidal analysis packages (e.g. Bell et al., 1996), even though Figure 3 shows that it is resolvable at higher latitudes, at least in BPR data. One supposes that the reason for lack of interest in this constituent by previous tidal analysts has been due to its

smaller amplitude at low- and mid-latitudes and to the generally higher level of noise in tide gauge records.

There are several complications we are aware of in the above analyses. One is that, when measurements are combined from different locations, then the observed tidal amplitudes should be adjusted for spatial-variations in water density, latitude-dependent variations in acceleration due to gravity, and depth-dependent compressibility of sea water. However, these will be at the ~1% level (Ray, 2013) and so are much less than other uncertainties.

A second complication concerns whether imperfections in our tidal analyses and subsequent averaging of the BP residuals into daily means of BP could have aliased residual components of the main diurnal and semi-diurnal tides into frequencies similar to those of the three long-period tides. We do not believe this is an important issue. All of the tidal analyses were subjected to quality control to check that tidal and non-tidal components of the records were separated efficiently. However, any residual tidal signals would then have been considerably reduced by the daily averaging. For

example, the amplitude of any residual M2 would have been reduced to approximately 3.5% of its original value and aliased into the period of MSf (14.77 days). Consequently, while it is possible that aliasing could have contributed to some of the MSf in Figure 3(a), we believe most of that to be real. Furthermore, it is hard to see how the observed Mf, Mm and Mt could have been affected to any significant extent by aliasing. In principle, residuals of the tiny constituents OP2, Lambda2 and SNK2 could be aliased into Mf, Mm and Mt respectively, although reduced to negligibility by the daily averaging. Lambda2

is included explicitly in the tidal analysis. The other two are interaction constituents (see below) and do not appear as significant lines in the tidal potential (Cartwright and Tayler, 1971).

A further complication is that there will be other constituents present in the data (i.e. genuine and not-aliased ones) with a similar period to Mf, Mm or Mt. We have ignored this complication for present purposes as the other constituents are likely to be small. In the case of Mf, the other main constituent will be MSf. MSf has an amplitude 9% of that of Mf in the

equilibrium tide, which is similar to that found in the composite BPR record (Figure 3a). Similarly, MSm (period 31.81 days) has an amplitude of 19% of Mm in the equilibrium tide, and MSt (period 9.56 days) is 19% of Mt. However, there is little evidence for significant amounts of either in Figure 3(a). Again, these other constituents should be separable from Mf, Mm and Mt given a year of data. One might imagine a more sophisticated harmonic expansion in future work in which information on these and other constituents is inferred from ocean models.

Another complication is that observations of the three long-period tides considered here can contain contributions from nonlinear interactions between shorter-period tides. For example, the difference between K1 and O1 frequencies is identical to that of Mf, and so their interaction can contribute to the observed Mf. K2 and M2 interactions can also contribute. Similarly, M2 and S2 can provide an interaction with the same frequency as MSf, which is similar to that of Mf. N2 and M2 interaction can contribute to Mm. An interaction will have an $f$ and $u$ determined by the product of the





individual $f$ and $u$ values of the two short-period tides involved (see Table 4.4 of Pugh and Woodworth, 2014). Therefore, interaction nodal factors will be different to those of the long-period tide. This complication is primarily an issue for shallow waters, rather than the deeper ocean areas of the Drake Passage where our BPR measurements are located. Nevertheless, it should be possible to estimate the contributions from such interactions using tide modelling.

A final complication relates to all BP spectra having a continuous non-tidal background in addition to a tidal line spectrum (e.g. Figure 3). The background will tend to increase the amplitudes calculated for each tidal constituent (see Appendix B of Munk and Cartwright, 1966 and discussion in Wunsch, 1967). We simply note that this aspect would impact primarily on our determination of Mm. Another issue to do with the background is that it is not white noise. As mentioned above, this could lead to the errors in the harmonic analysis regressions being underestimated (e.g. Williams, 2003). In fact,

the background spectra for all of the 45 BPR deployments are similar and can be parameterised reasonably well by a (frequency [k]) dependence where k ~ -1.5 (Supplementary Figure 3). This suggests similar biases in estimated errors for each constituent for each deployment. Such biases, as long as they are similar in each case, should not significantly affect the fits to determined parameters from all deployments in Figures 5-7.

**5 Conclusions**

If one has several decades (or at least 19 years) of good tide gauge (or, in theory, BPR) data available for a tidal analysis then, if background noise levels allow, it should be possible to avoid having to consider the nodal sidebands as perturbations of the main harmonic via the use of 'nodal factors' $f$ and $u$. Instead, one can treat them as independent constituents and make an explicit determination of their amplitudes and phase lags. Examples of such analyses of long records include those

of Amin (1983) and Foreman and Neufeld (1991).

However, in practice most tidal analyses are made using one or several years of data, for which assumptions are required for $f$ and $u$. The drawbacks of this approach have been recognised for many years, but primarily for the semidiurnal and diurnal constituents. As far as we know, the question of whether the variation of the long-period tides through the nodal cycle differs from equilibrium expectations has never been investigated properly.

In this paper, we have used data from 45 separate BPR deployments in the Drake Passage, and 31 years of hourly tide gauge data from the Vernadsky station in Antarctica, to estimate how well the nodal variation of the amplitudes and phase lags of Mf, Mm and Mt compare to expectations from the equilibrium tide. Our analysis uses simple harmonic expansions of daily values of BP or sea level at each location.

The combined data set provides information on how the amplitudes and phase lags of each constituent vary between

the north and south sides of the Passage. The measurements indicate that amplitudes are similar throughout the region, which is consistent with a state-of-the-art ocean tide model (FES2014, 2018). Phase lags for Mf and Mt are ~20° larger in the south than in the north, which is also consistent with the model. However, the observed south-north difference in Mm phase lag is consistent with zero, compared to ~10° in the model. In fact, the Mm difference is probably consistent with the model given the uncertainties, and at least the BPR data and FES2014 are in agreement on indicating a smaller meridional gradient for





Mm phase lag than for the other two constituents. Any detailed differences for all the long-period tides may be understood better by future modelling.

However, our main interest is in the temporal variability of the long-period tides. The variation of the amplitudes and phase lags of Mf and Mt in the BPR data have been found to be consistent with those suggested by the equilibrium tide within their uncertainties. To a great extent this is an expected finding given that, as explained in the Introduction, the long-period tides are closer to equilibrium than the diurnal and semi-diurnal tides and the frequencies of the nodal sidebands are close to that of the central line. Nevertheless, this is a reassuring finding for tidal analysts who might now (in this region at least) be able to employ $f$ and $u$ for the long-period tides as anticipated. The variation in phase lag of Mm (or rather its non-variation) is also consistent with equilibrium expectations. The absence of an expected 13% variation in the amplitude of Mm (Equation 8) at 3-sigma level (or possibly less if, as explained above, our uncertainties were slightly underestimated) is probably due to the background of non-tidal variability in the ocean circulation in this energetic area and/or in our inability to account adequately for spatial variations in Mm amplitude with the use of FES2014.

Our study has shown clearly that BPR data have many advantages over conventional tide gauge measurements in long-period tidal studies such as this. Nevertheless, it is the case that there is a lot more tide gauge data available for study worldwide than BPR data (Woodworth et al., 2017). Therefore, an obvious recommendation following from the present work is that tide gauge data be investigated more completely in order to investigate whether the temporal variation of long-period tides conforms to equilibrium expectations, perhaps by employing 'stacks' of records, as has been used previously to investigate other long-period components of tide gauge records (e.g. Trupin and Wahr, 1990).

**Competing interests**

The authors declare that they have no conflict of interest.

*Acknowledgements.* The programme of Drake Passage bottom pressure measurements was led by scientists from the National Oceanography Centre including Ian Vassie, Mike Meredith, Chris Hughes and Miguel Ángel Morales Maqueda. The bottom pressure recorders were designed, constructed and deployed by Bob Spencer, Peter Foden, Jeff Pugh, Steve Mack, Geoff Hargreaves and others. The help of the British Antarctic Survey with the deployments is much appreciated. Data sets were processed by David Blackman and Philip Axe. The NERC 'Weighing the Ocean' project (Grant No. NEW344022) was led by Mark Tamisiea and Chris Hughes. Loren Carrère is thanked for advice on the FES2014 ocean tide model which is distributed by AVISO (http://www.aviso.altimetry.fr/). NEMO ocean model data and advice on this study were provided by Chris Hughes. We are grateful to Richard Ray and Simon Williams for comments on aspects of this work. Part of this work was funded by UK Natural Environment Research Council National Capability funding. Some figures were generated using the Generic Mapping Tools (Wessel and Smith, 1998).



**Appendix: The accuracy of Doodson's nodal factors**

The formulae for $f$ and $u$ presented in Doodson (1928) and Doodson and Warburg (1941) are more complicated than those in Equations 3, 4 or 8, in that they include additional terms depending on the cosines and sines of $2N$ and $3N$. However, the

ones we have used are adequate for the present paper. It is useful to explain where they come from.

Imagine a constituent of unit amplitude described schematically by $cos(\omega t)$, where for simplicity we have ignored the $A$ and $G$ in Equation 2. Consider the constituent as having a single important nodal sideband with an amplitude $R$ which is less than 1, and an angular frequency $\omega + n$, where $n = \left(\frac{2\pi}{18.61\ years}\right)$ is the angular frequency of the nodal angle $N' = -N$ (Doodson, 1921). This $\omega + n$ situation represents Mf and its sidebands. Mt and K2 can be represented similarly. M2 has its

single important sideband at $\omega - n$. Although most lunar constituents have one sideband that is much larger than the other, there are some constituents for which the amplitudes of the sidebands are approximately the same, such as Mm, see below.

Therefore, in the example of Mf, we can express the total tide as:

$$cos(wt) + Rcos\big((w + n)t\big) = [1 + Rcos(nt)]\,cos(wt) - [Rsin(nt)]sin(wt)$$

[A1]

The nodal factor for amplitude ($f$) can then be expressed by:

$$f^2 = 1 + R^2 cos(nt)^2 + 2Rcos(nt) + R^2 sin(nt)^2 = 1 + R^2 + 2Rcos(nt)$$

$$f = \sqrt{1 + R^2}\sqrt{1 + \frac{2Rcos(nt)}{1 + R^2}}$$

                                                                                                                         [A2]

We can expand the second square-root by a Maclaurin series:

$$f = \sqrt{1 + R^2}\left[1 + \left(\frac{2R}{1+R^2}\right)\frac{cos(nt)}{2} - \left(\frac{2R}{1+R^2}\right)^2 \frac{cos(nt)^2}{8}\ etc.\right]$$

[A3]

from which the second term provides the nodal time dependence of $f$ i.e. $\frac{Rcos(nt)}{\sqrt{1+R^2}}$. When $R$ is very small this is simply $Rcos(nt)$. (In the case of M2, for which the sideband is at $\omega - n$, then it becomes $-Rcos(nt) = -0.037\ cos(nt)$, as in Equation 3.) However, the main sideband of Mf has a much larger $R$ value of 0.414, from which Equation A3 gives a time dependence of $0.382\ cos(nt)$. As can seen from Equation 4, Doodson ignored the complication of the denominator and

taken $Rcos(nt)$ to apply for Mf also.



The first and third terms provide the time-independent part of $f$ for which Doodson took the time-average value of the third term. When $R$ is very small, then the sum of the first and third terms can be approximated by:

$$\left(1 + \frac{R^2}{2}\right) - \left(\frac{R^2}{4}\right) = 1 + \frac{R^2}{4}$$

[A4]

from which one obtains 1.0004 for M2 (Doodson, 1928). When $R$ is larger, we would have:

$$\sqrt{1 + R^2} - \frac{R^2}{4(1 + R^2)^{\frac{3}{2}}}$$

[A5]

which gives a value for Mf of 1.0485 given that $R = 0.414$. However, once again, Doodson appears to have assumed the small $R$ approximation of Equation A4, giving the 1.043 in Equation 4.

From Equations 2 and A1, we can express the nodal factor for phase lag as:

$$u = tan^{-1}\left(\frac{Rsin(nt)}{1 + Rcos(nt)}\right)$$

[A6]

and from the Maclaurin series $tan^{-1}x = x - \frac{x^3}{3} + \frac{x^5}{5}$ $etc.$ for $-1 < x < 1$, this gives $u = Rsin(nt)$ if the denominator is taken to be 1.0 for small values of $R$. Once again, this is clearly an acceptable approximation for M2. However, Doodson also used this approximation for Mf, resulting in the $u = 0.414\,sin(nt)$ radians or $23.7°\,sin(nt) = -23.7°\,sin(N)$ as in Equation 4.

As a test of whether these approximations by Doodson matter, Figure 9(a,b) shows the values of $f$ and $u$ that one obtains for Mf by calculating them rigorously using Equations A2 and A6, or by using Doodson's formulae. It can be seen that Doodson's values of $f$ and $u$ are good approximations, with standard deviations of the differences between the red and blue curves of 0.03 and 3.6° respectively. Therefore, they can be adopted reliably for analysis of generally noisy tide gauge or BPR data. However, in other tidal applications they may not be adequate. For example, Ray and Egbert (2012) made a study of fortnightly variations in earth rotation. When the nodal sidebands of Mf were treated rigorously, and additional double-nodal and double-perigean sidebands were included (i.e. sidebands with angular speeds which differ from that of the main line by the angular speeds of $2N'$ and $2p$ respectively, where $p$ is the angle of lunar perigee), then improvements were obtained over the Doodson descriptions of $f$ and $u$ we have used here, which in turn improved upon their interpretation of high-precision length of day information.



As mentioned above, the formulae for $f$ and $u$ presented in Doodson (1928) and Doodson and Warburg (1941) are more complicated than the simplified ones discussed here. For example, his values for Mf include the double-nodal terms considered by Ray and Egbert (2012) (but not the double-perigean ones), and these more complete expressions will have been included in most tidal analysis and prediction software packages.

Finally, we can refer to Mm which has two nodal sidebands with amplitudes that are the same to within 1%, and that have opposite sign to that of the Mm central line in the harmonic expansion of the tidal potential (Cartwright and Tayler, 1971). The total tide can then be expressed as:

$$cos(wt) - Rcos\big((w+n)t\big) - Rcos\big((w-n)t\big) = [1 - 2Rcos(nt)]\,cos(wt)$$

[A8]

It is straightforward to see that in this case when $R = 0.065$ that $f = 1 - 0.130\,cos(N)$ and $u = 0.0$ as shown in Equation 8.

A more complicated discussion of Mm would include its other sidebands. Mm has Doodson number 0,1,0,-1,0,0. Its main double-perigean sideband 0,1,0,1,0,0 (i.e. differing by $2p$ from the main line) has an amplitude ~5% of Mm itself (as does the double-perigean sideband of Mf), while there is a component 0,1,0,1,1,0 (i.e. differing by $2p + N'$ from the main line). There is even a $3^{rd}$-degree single-perigean component 0,1,0,0,0,0 (i.e. differing by $p$ from the main line). The overall nodal factors $f$ and $u$ can then be obtained via:

$$fcos(u) = 1.0 - 0.130\,cos(N) - 0.0535\,cos(2p) - 0.0216\,cos(2p - N) - 0.0551\,sin(p)$$

$$fsin(u) = -0.0535\,sin(2p) - 0.0216\,sin(2p - N) + 0.0551\,cos(p)$$

[A9]

where the amplitudes of each term are taken from Cartwright and Tayler (1971) and that of the $3^{rd}$-degree term is evaluated at 58°S. Figure 10 indicates the simple nodal components of $f$ and $u$ as described by Equation 8 (or A8) by thin black and blue lines respectively. The overall values after combining all components in Equation A9 are shown by the thick lines. (This would pre-suppose that both $2^{nd}$ and $3^{rd}$-degree long-period tides have a near-equilibrium behaviour. The overall values if one were to include only $2^{nd}$-degree components are shown in Supplementary Figure 4.) Equation 8 can be seen to be a good approximation of the overall $f$ and $u$ in spite of the other sidebands.





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

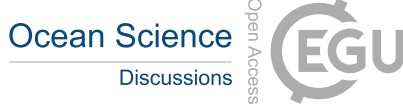

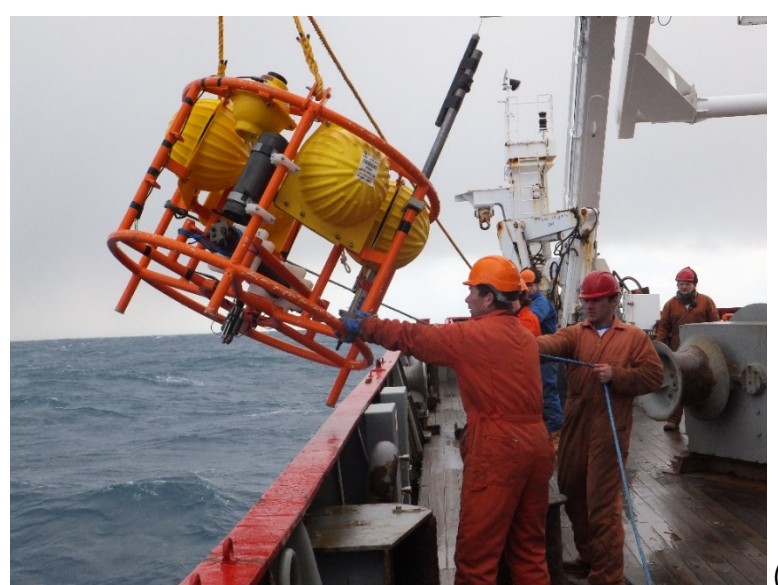

(a)

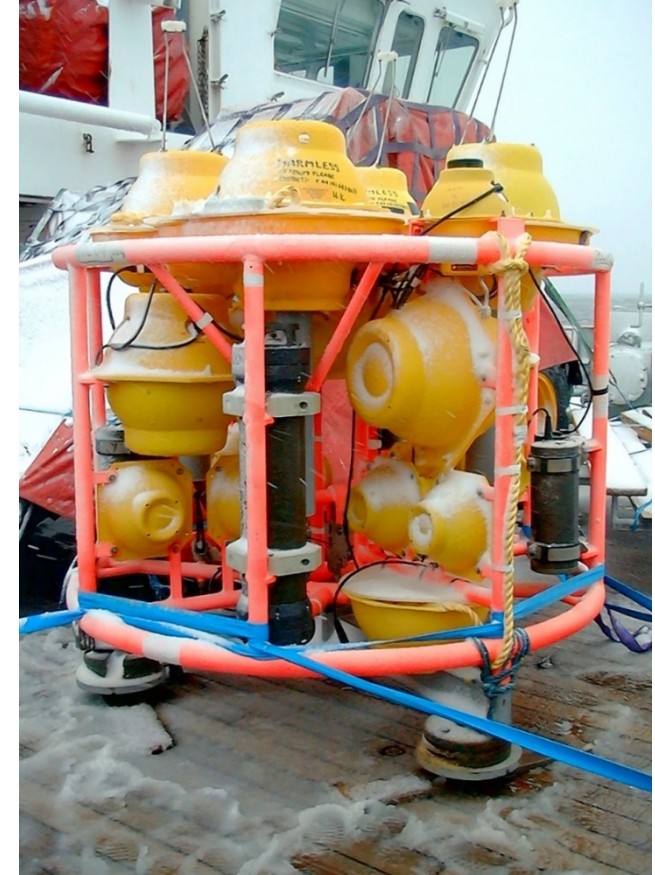

(b)





(**previous page**) **Figure 1.** (a) The 'Mk.IV' and (b) MYRTLE bottom pressure recorders used in the Drake Passage. (a) shows the lander being recovered. Therefore, it is without its heavy ballast frame on which the orange lander frame sits when deployed on the sea bed. A ballast frame can be seen in (b). Pressure transducers are installed in the horizontal logger tube in (a) and vertical tube in (b). Photographs from the National Oceanography Centre.

**Figure 2.** Map of the Drake Passage showing the locations of the 45 BPR deployments and the Vernadsky (Faraday) research station. Red dots indicate deployments by bottom landers based on the 'Mk.IV' design, while green stars indicate deployments by MYRTLE instruments. Depths of 1000 and 3000m are shown by the black and blue contours respectively.





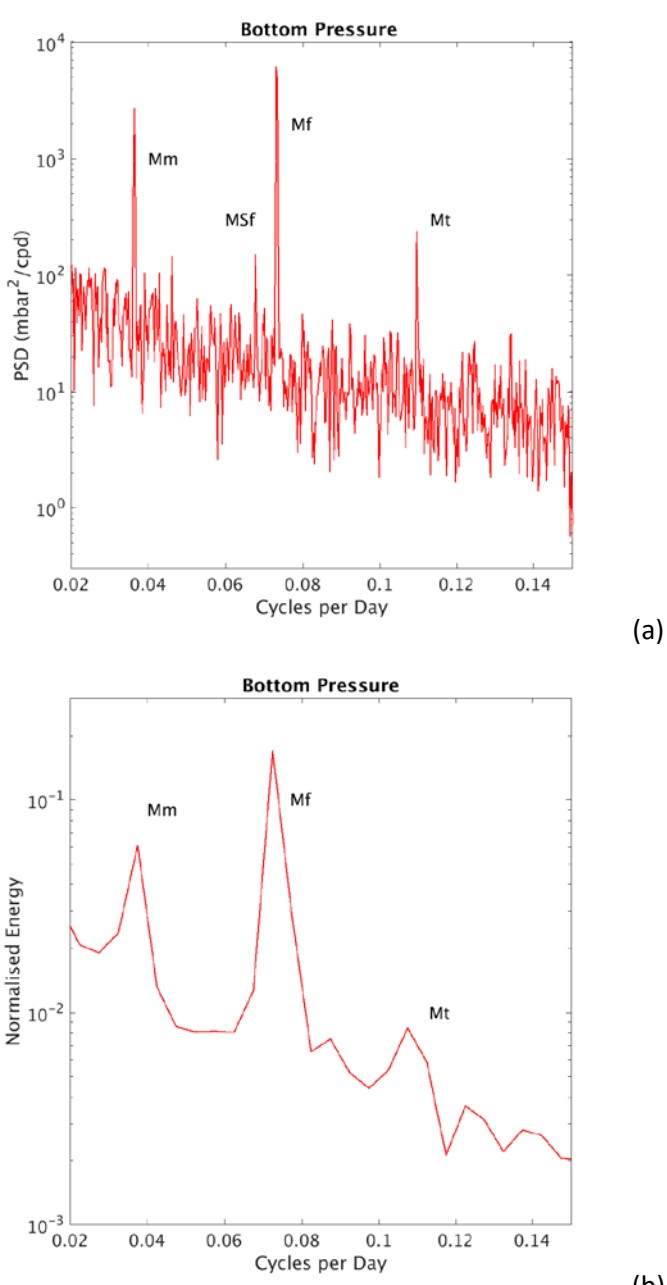

(a)

(b)

**Figure 3.** (a) Power spectral density (PSD) of a composite, continuous record of daily mean BP in the Drake Passage spanning over 26 years, (b) An averaged, normalised power spectrum typical of each record made from 20 of the 45 BPR time series which had no data gaps for which the standard deviation of the regression fit in terms of 3 harmonics was at least 50% of that of the times series itself, so ensuring a significant tidal component (also see text).





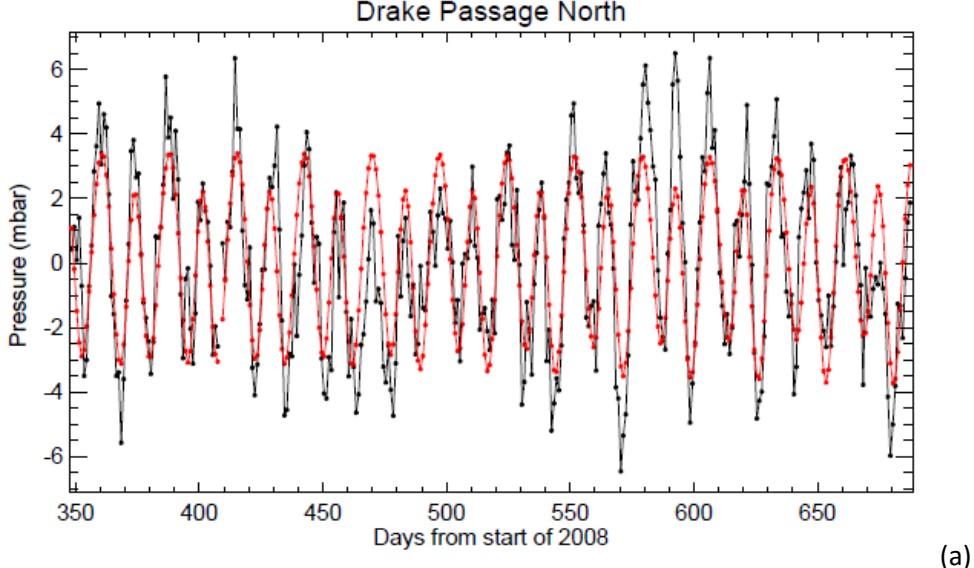

(a)

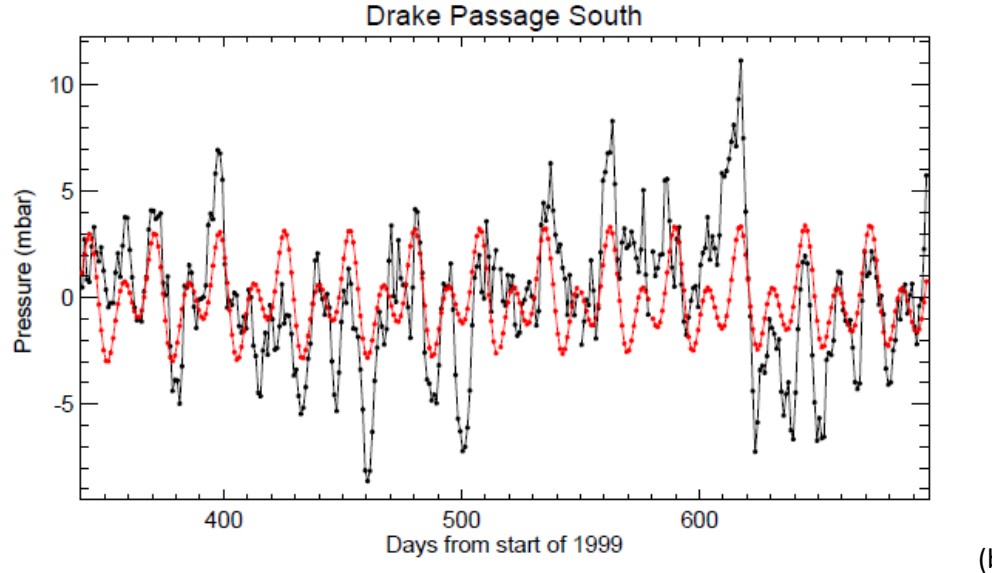

(b)

**Figure 4.** (a) An example of a fit (red) to daily mean BP values from the north side of the Drake Passage (black) in terms of
three harmonics with the frequencies of Mf, Mm and Mt during a period when the amplitude of Mf was larger than average,
(b) An example of a record from the south side of the Drake Passage during a period when the amplitude of Mf was smaller
than average and so the contribution of Mm is more apparent.



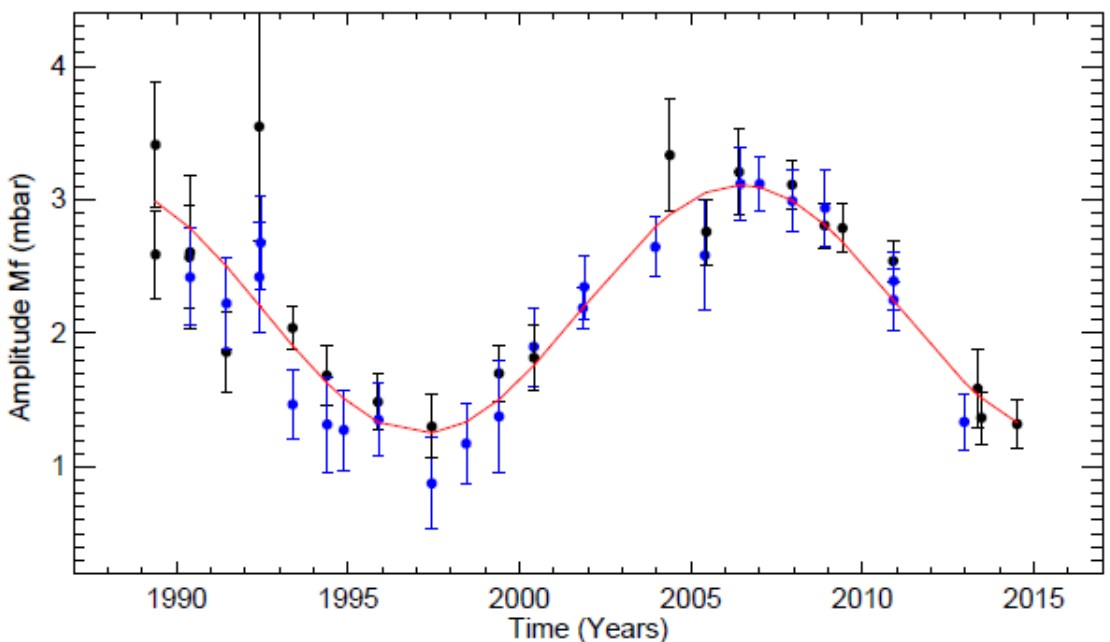

Figure 5(a)

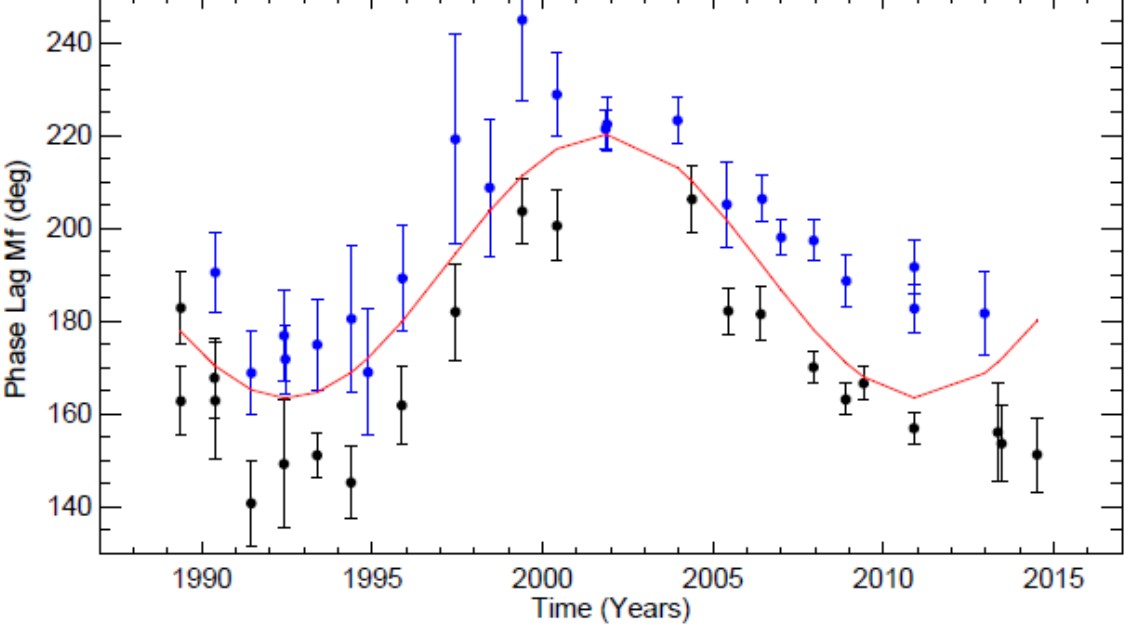

Figure 5(b)





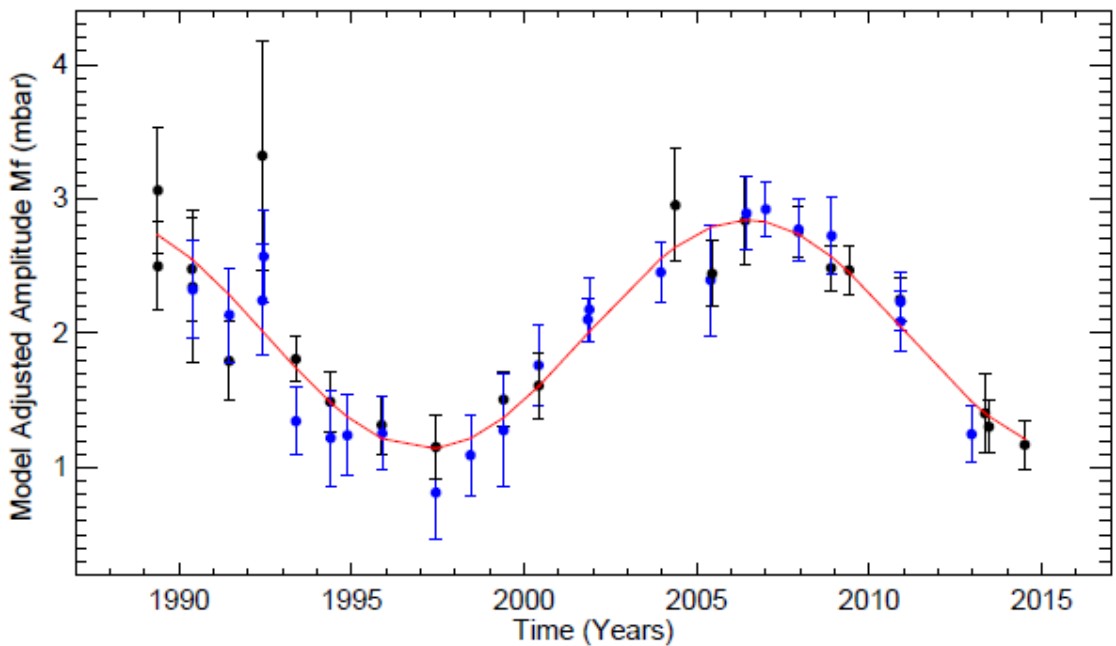

Figure 5(c)

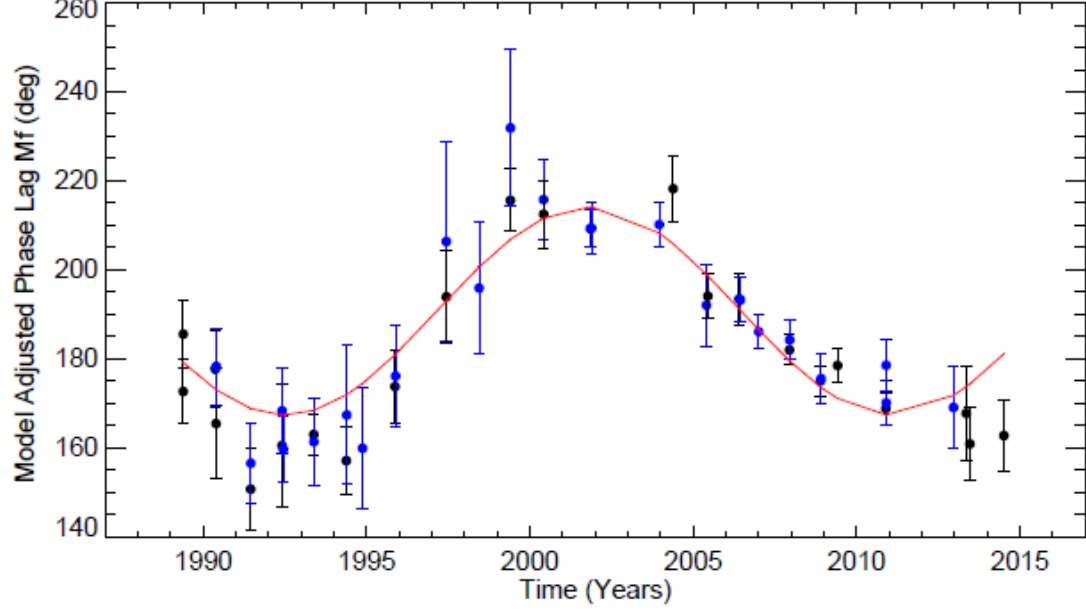

5          Figure 5(d)





**(previous two pages) Figure 5.** (a) Amplitude (mbar) and (b) phase lag (degrees) of Mf obtained from each BPR deployment plotted versus the central date of the record. Error bars show one standard error. Black and blue points indicate deployments north and south of 58 °S respectively. Red lines indicate fits with nodal variations as described in the text. (c,d) As (a,b) but with adjustments using the FES2014 model.



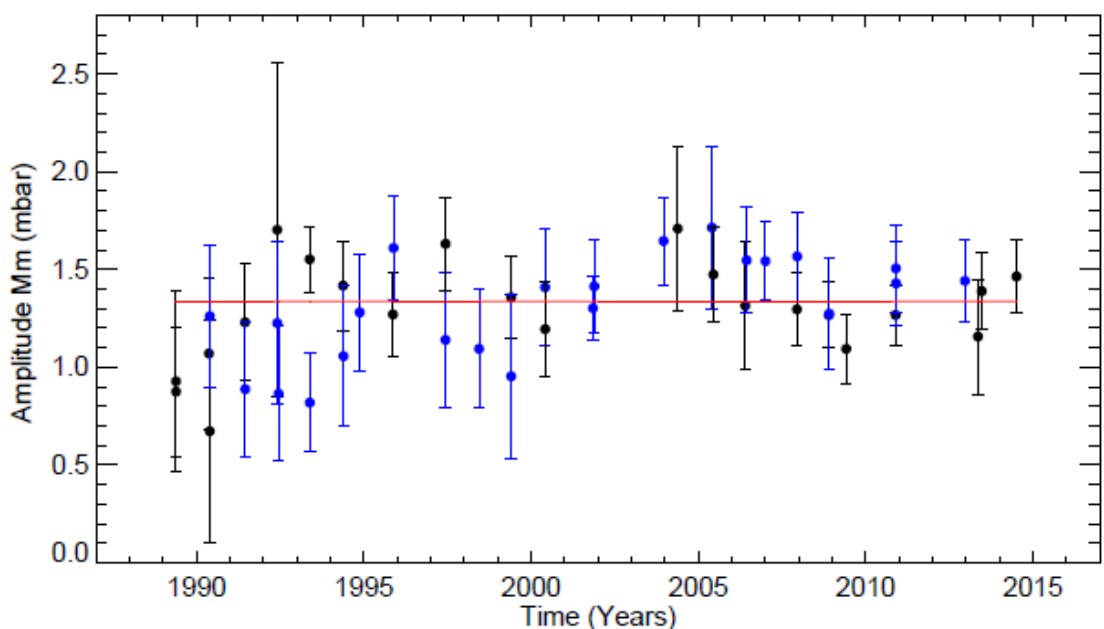

Figure 6(a)

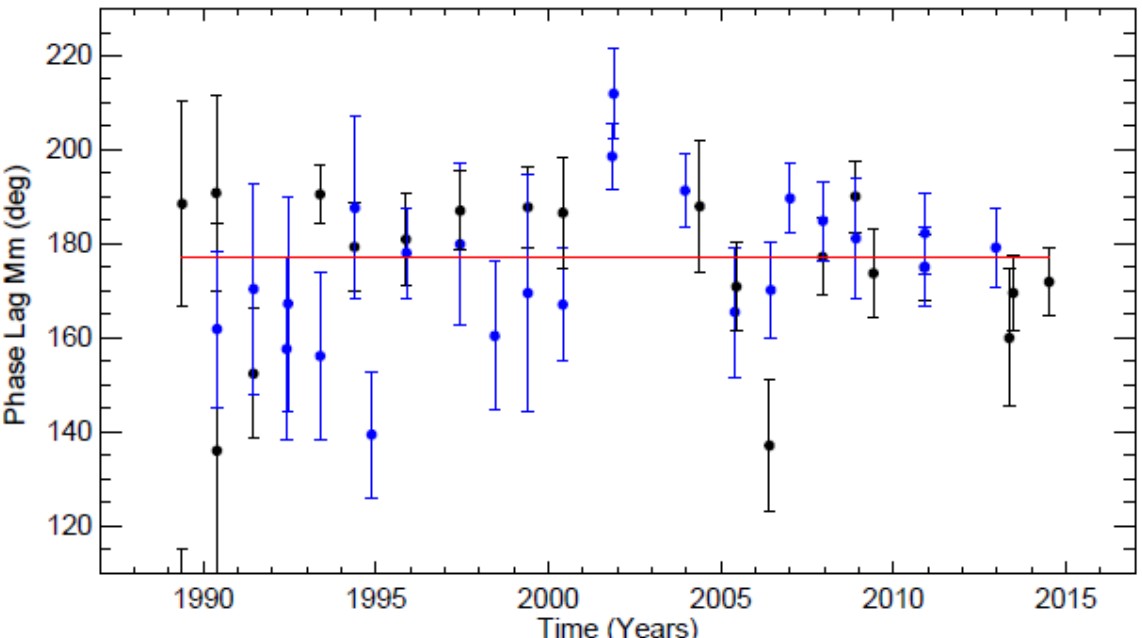

Figure 6(b)




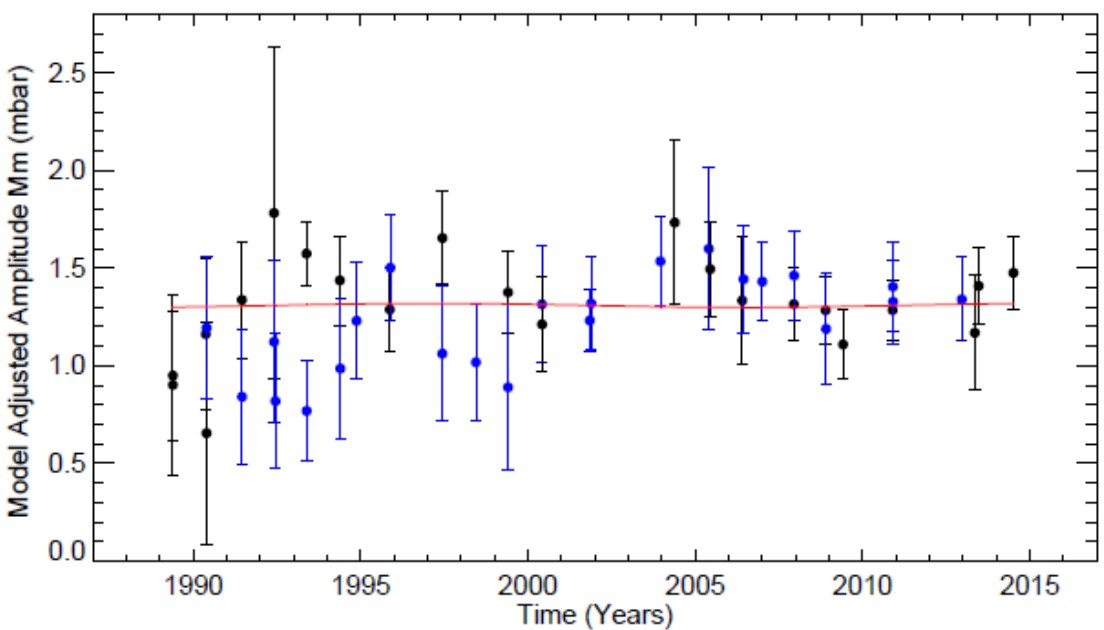

Figure 6(c)

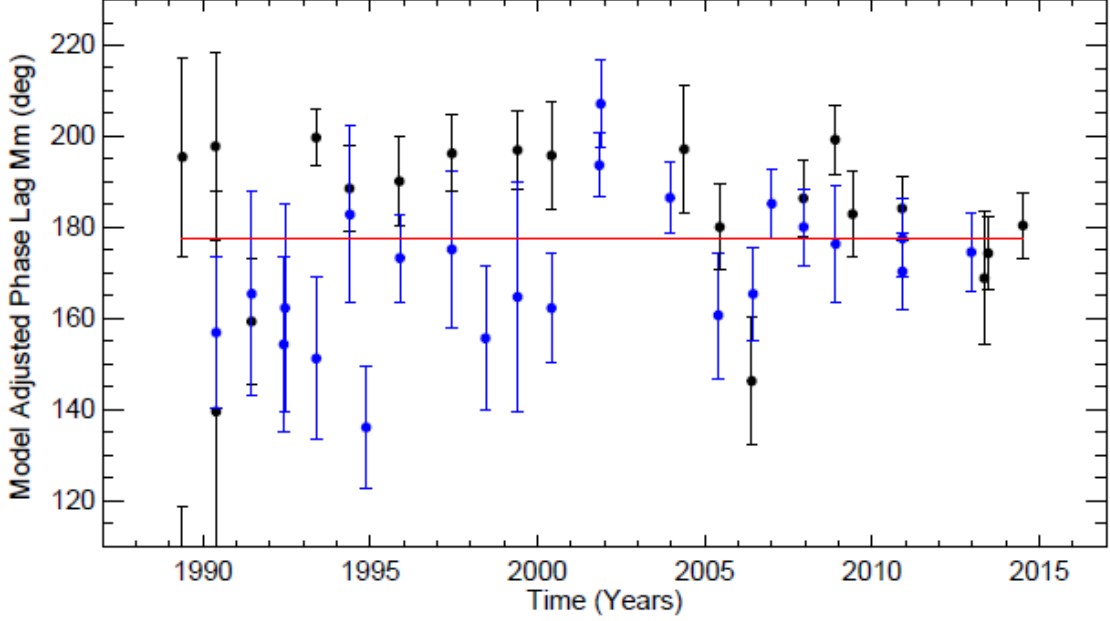

Figure 6(d)





**(previous two pages) Figure 6.** (a,b) As Figure 5(a,b) but for the Mm long-period tide. The red line in (b) indicates the median phase lag instead of a nodal fit. (c,d) As (a,b) but with adjustments for different deployment locations using the FES2014 model.


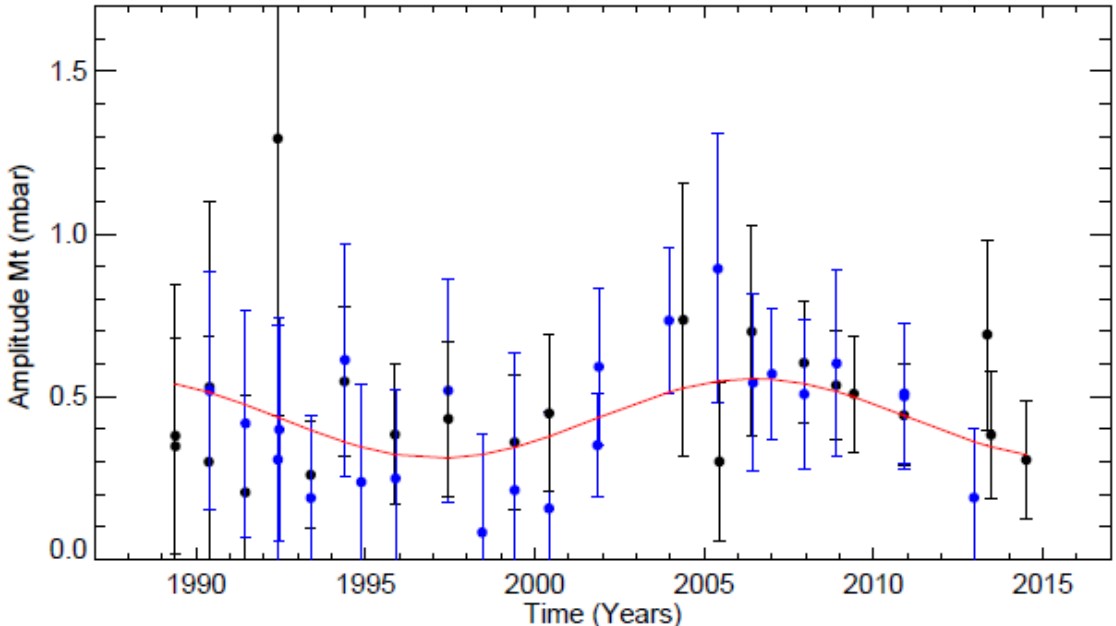

Figure 7(a)

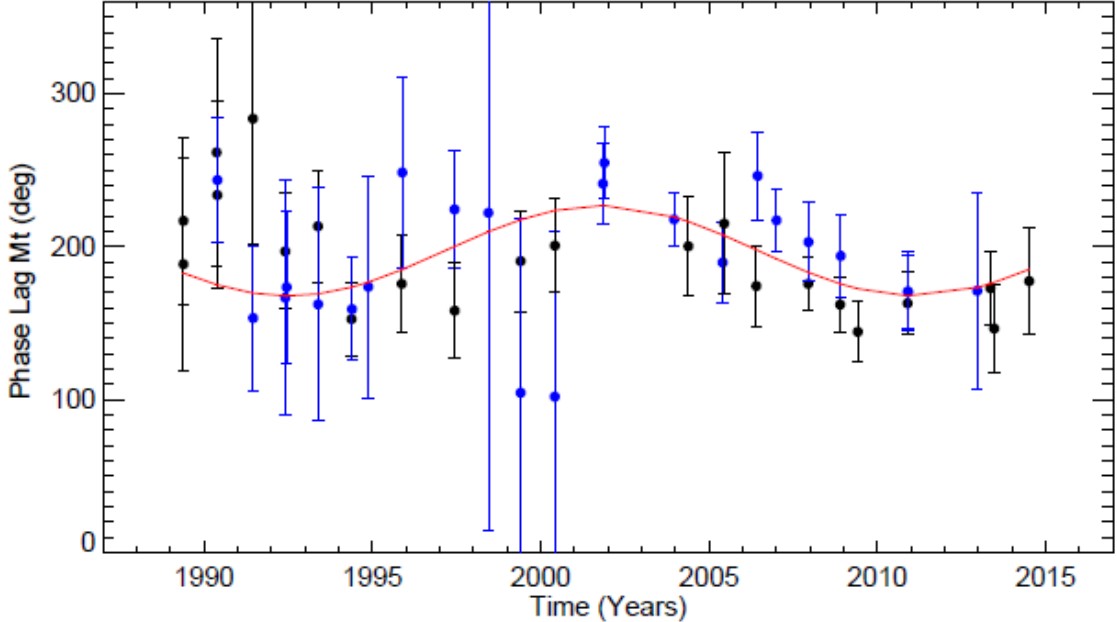

5   Figure 7(b)

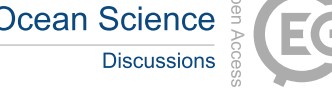



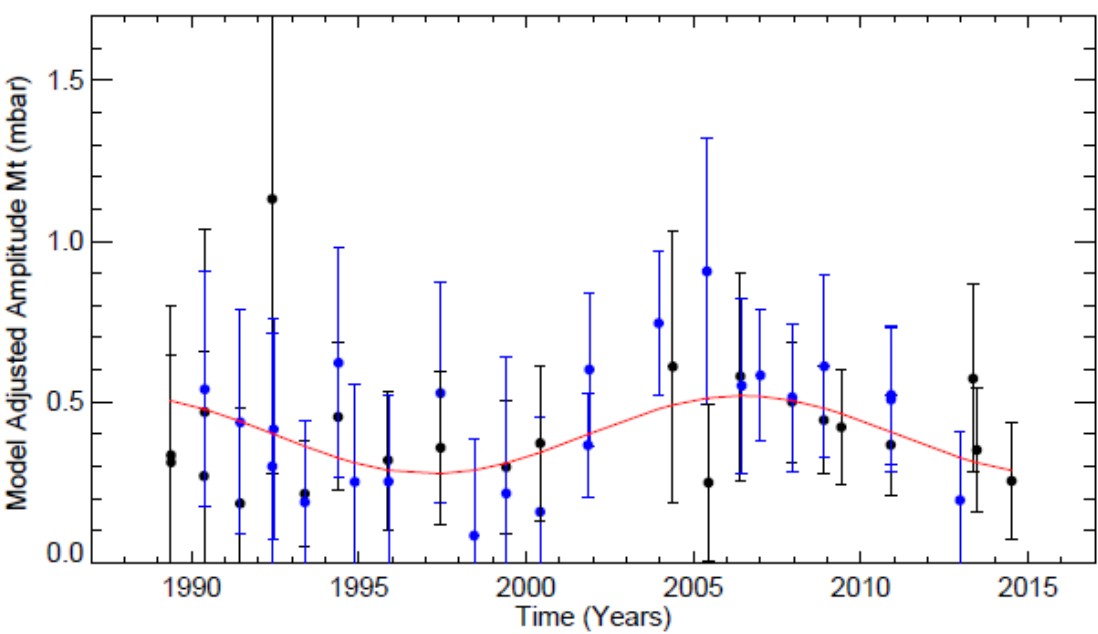

Figure 7(c)

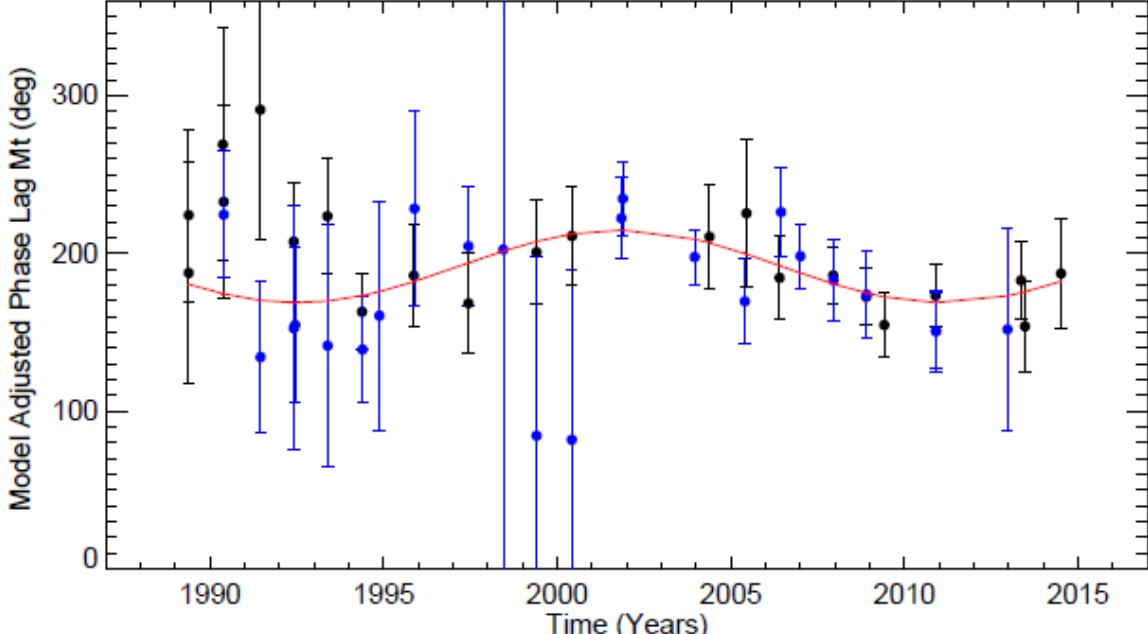

Figure 7(d)



**(previous two pages) Figure 7.** (a,b) As Figure 5(a,b) but for the Mt long-period tide. (c,d) As (a,b) but with adjustments using the FES2014 model.



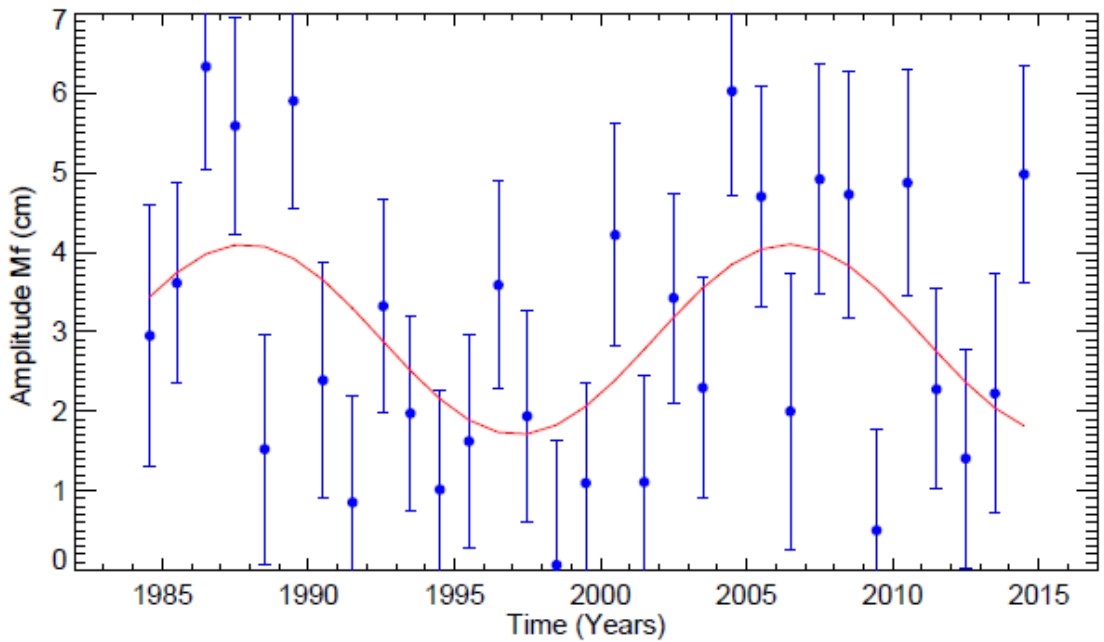

(a)

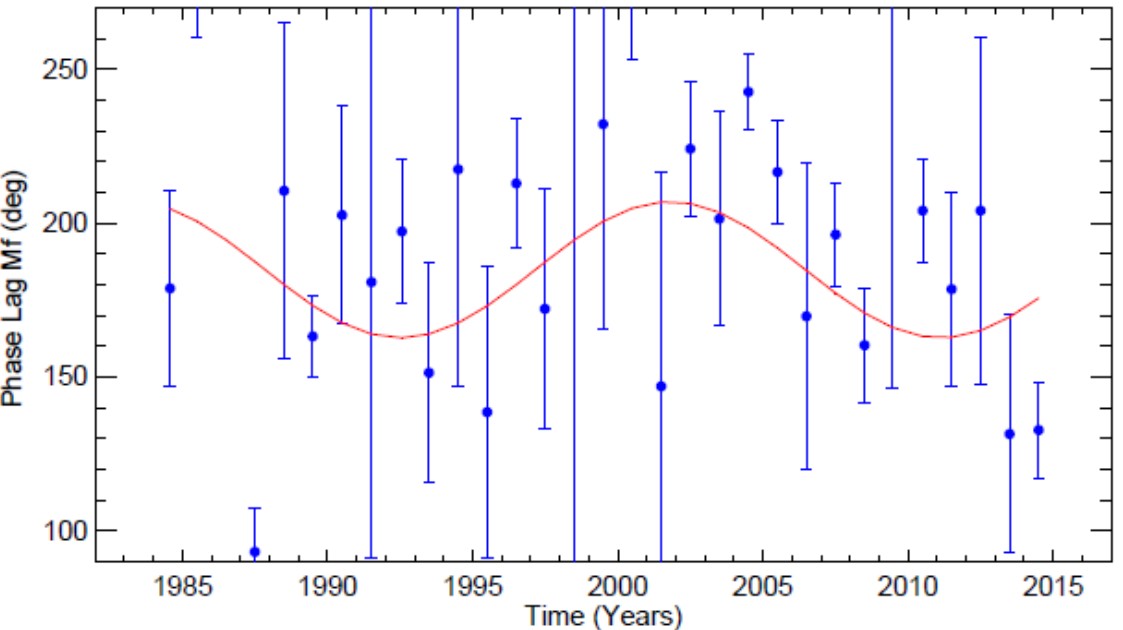

(b)

**Figure 8.** (a) Amplitude (cm) and (b) phase lag (degrees) of Mf obtained from Vernadsky tide gauge data.







**Figure 9.** Values of (a) $f$ and (b) $u$ for Mf computed by Equations A2 and A6 (red) or using the Doodson parameterisations
5    (blue) as a function of $N' = -N$, $N$ being the longitude of the lunar ascending node.




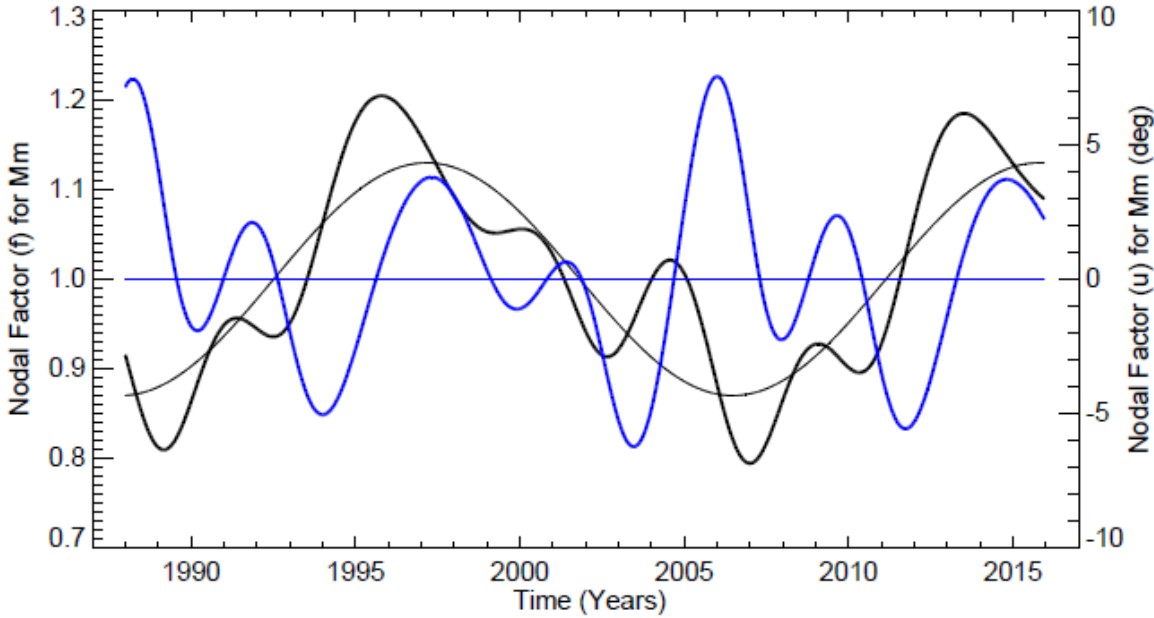

**Figure 10.** The nodal factors $f$ and $u$ for Mm at 58°S. The two approximately-equal nodal sidebands result in the $f$ and $u$ in Equation 8 (or A8), as indicated by the thin black and blue lines respectively, with values shown on the left and right axes respectively. The overall values of $f$ and $u$ for Mm, after taking into account the sidebands included in Equation A9, are shown by the thick black and blue lines respectively.