# Peer review of "The nodal dependence of long-period ocean tides in the Drake Passage"

_Ocean Science, 2018_

## Referee Comment (RC2) · X. Feng (Referee) · 29 May 2018

This paper is to confirm the capability of the equilibrium tide theory in predicting the nodal (18.61 years) variations of long-period tides at high latitudes, using ~31 years of bottom pressure recorder (BPR) data. Due to the fact that the long-period tides are usually very small tidal signals compared to the dominant diurnal and semi-diurnal constituents such as O1 and M2, detecting their long-term variations (i.e.nodal cycle) is always a challenge for oceanographers. It's good to see Woodworth and Hibbert making progress on this topic thanks to their very careful analysis work. It is more interesting to see this paper tackling the nodal cycle of long-period tides in the ACC region where ocean current is energetic and is supposed to have strong influence on these tidal signals. In addition, the diagnostic approach in this paper can be easily

extended to other regions, which will help to complete the world map of such tidal analysis. This is an interesting paper and is pleasant to read. I only have some suggestions as below.

1. P1 31: 't=0,' =>'t=0. Eq[1] is further modified to;'?

2. P6 6-8: Are the daily mean values suitable for resolving Mt (9.13 days)? Using daily means might be one of reasons for large error bars in Fig7. Though there are some discussions on the complications of this analysis approach, it will be better if authors can further discuss(/investigate) the dependence of long-period tides, especially Mt, on data temporal resolution.

3. P7 4-7, Fig4 and other places in text: why is non-tidal variability larger in the south side as however there are more eddies in the north as authors mentioned in P12 5? MJO (intraseasonal) is taken as one potential contributor but it seems to me that there are still some significant features at longer timescales (e.g. in Fig4b between 350day and 450day, between 510day and 610 day). A bit more explanations/speculations are suggested to add here.

4. Subsection 3.2: It compares the long-period tides derived from BPR and also the tide gauge record. How the power spectral distributions differ between those two kinds of records? If BPR has advantages of resolving long-period tides over tide gauge data, due to less non-tidal variability, one may expect there are more noises close to the 3 constituents' frequencies (Fig3). Is this true? It's good to show this merit.

5. P8 1-3 and Fig5b: It's worth proposing some explanations why such north-south differences are observed here, when this is not expected from the theory.

6. Fig 6&7: small amplitudes and large error bars make it difficult to detect the nodal cycle. It seems to me that error bars are slightly larger in the 1st decade. Is this related to BPR data density used? It'll be good if the data availability (after QC) of such 45 BPR records is provided.

7. Author used FES2014 model to discuss the spatial variation of long-period tide parameters. FES models are assimilated by satellite altimeter data, which to me however have some limitations at high latitudes. Is this a noticeable concern here?

---

## Author Response (AR1)

Joseph Proudman Building 6 Brownlow Street Liverpool Merseyside, L3 5DA United Kingdom Tel: +44 (0) 151 795 4800 Fax: +44 (0) 151 795 4801

www.noc.ac.uk

Editor, Ocean Science

26 June 2018

Dear Sir,

Paper Submitted to Ocean Science

Many thanks for the reviewer comments on our paper submitted to Ocean Science, which appeared in OS Discussions, entitled "The Nodal Dependence of Long-Period Ocean Tides in the Drake Passage" (os-2018-50). They were very useful. I have uploaded our replies to the comments and these are included again below

We have made a number of small changes to the text following the comments, and a few others of our own. The only major change has been to Section 3.2 which discusses Vernadsky data. That has been rewritten and expanded, and there are extra figures to do with that (Figures 3c,d and 8c,h). We hope there are not now too many figures, but most of them can be printed fairly small. In addition, we have added a Table 1 to summarise the many numbers given in the text.

Also below is a Word comparison file which shows where changes have been made. The Supplementary Information file is not changed.

If there are problems, I can be contacted at plw@noc.ac.uk.

Many thanks for your help with this.

Yours sincerely

Philip L. Woodworth

Philip Woodworth

**Reviewer 1**

We are grateful for the time that both reviewers spent on this paper. The comments of Reviewer 1 are given below followed by our replies. The page and line numbers refer to the version in OS Discussions.

P2 131: maybe add the reference Lyard et al 2006

Done

P4 125: OK, but in this high latitude region, the ocean response to atmospheric pressure can be significantly different from IB + effects of wind not negligible => might need to use a model forced by the atmosphere (at least a barotropic model for high frequencies) to remove correctly this non tidal variability. Have you done this test ?

We are simply saying here that sea level variability due to air pressure changes are largely compensated for in the BP records by air pressure itself, at the timescales we are interested in, thanks to the inverse barometer. That automatically removes a lot of the non-tidal variability from the records. We believe that is all correct. We agree that there are still other processes in the ocean (caused by the wind or whatever) that result in non-tidal variability – those are investigated using the NEMO model data discussed on page 12 which the reviewer commented on (so, yes, the 'test was done').

P4 129: "low-frequency process" : what are the frequencies concerned ? annual/semi annual only or some other components ?

This was badly worded. Instead of 'primarily a low-frequency process' we now say 'a slow monotonic process'. We have also added an extra reference (Polster et al., 2009) and the three references now given should be adequate to give the reader an idea of the problems of drift in deep pressure sensor data.

P6 17: have you considered the same length of record for each BPR ? if not, can you estimate the impact of the different lengths of record on the harmonic estimation of Mf, Mm, Mt ? this impact is likely not negligible and should be considered in the discussion.

This is related to the question for page 11 line 15, see below. Almost all deployments were annual ones apart from those that used the MYRTLE instruments. If the lengths had been very different, then we agree that the lengths and noise contents of each record would determine differently the size of the uncertainties for amplitude and phase (Eq 5). However, we cannot see that being a problem in itself. We have worked with the computed formal errors from each regression and a bigger issue with this approach, as the text discusses, is the assumption of white noise for the computed errors.

P6 129: sentence not clear. Please rephrase.

The sentence should now be clearer.

P7 115 : add ref to eq 4

The same references apply as for M2 in Eq [3]. We have referenced Doodson and Warburg (1941) here again.

P7 133: "28.4 +/-1.4\_" : what about the sign? Do you obtain the same sign as in eq 4 ?

It obviously has the same sign as in Eq 4, as can be seen from Figure 5(a). We have added a few words to say that.

P8 14: add a sentence like "this N-S difference is likely explained by the dynamic response of the ocean at this frequency": see the spatial patterns of FES2014 showed in supplementary materials.

We have not done this. This would be getting ahead of things. The spatial variations (dynamic response) are discussed in detail in the next section using the FES2014 model.

P8 115: add ref to eq 8

The same references apply as for M2 in Eq [3]. We have referenced Doodson and Warburg (1941) here again.

P8 121: have you tried to fit cos or sin?

Let us explain what we did again. A first step using the data from each deployment is to find the individual amplitudes and phase lags determined using Eq [5], as explained in Section 2. A second step is that the set of all the values of amplitude are parameterised as a cosine peaking at year 2006.5 when N=0.0, as explained in Section 3.1. For example, that is easy to see for Mf from Eq 4. In the case of Mm the cosine should in theory be upside-down because of the negative sign in Eq 8 – such an upside-down shape was indeed obtained for Mm in Figure 6(a) although its small amplitude meant that it looks almost like a straight line – see the text on page 8 line 16.

For phase lags we fit a sine constrained to be zero at 2006.5 instead of a cosine (see end of page 7) as you can see from Eq 4 that the nodal variation in the equilibrium tide for Mf or Mt is expected to vary like sin(N).

We don't understand why the reviewer points to page 8, line 21 in this context as that sentence makes it clear that there is expected to be no nodal variation in phase lag for Mm (Eq 8), and indeed there is no evident nodal variation in the data, so we just show in Figure 6(b) a straight line at the average value of  $177.3^{\circ}$ .

P8 130-31: mean value 0.43 is smaller than in eq 4. Please explain

We don't understand the comment. The 0.43 mbar is the mean value of Mt amplitude. If the reviewer is asking why this is smaller than the 1.043 in Eq. 4, then the latter is the nodal factor 'f' which is a dimensionless multiplier of the amplitude in Eq 2 (the lines following explain why that is not exactly 1.0 in this case). Eq 4 applies to both Mf and Mt to a good approximation by the way, the average amplitude of Mf is of course the larger.

P8 133: "which follows from the larger average amplitudes in the second half of the data" : not clear, please explain

To be more complete, the text says "which follows from the larger average amplitudes in the second half of the data (Figure 7a)". So please take a look at Figure 7a again – the nodal

variation means that the amplitudes tend to be larger in the second half. However, we agree there is maybe a problem with saying 'average amplitudes' which we have changed to 'larger amplitudes on average'.

P9 117: "individual uncertainties approximately five times larger than for the BPRs": how do you explain this point ?

Because the non-tidal background is much larger in a coastal tide gauge record due to air pressures and winds than in a BP record from deeper water – this is the reviewer's own point above referring to page 4 line 25. See the next two points regarding spectra and DAC corrections to the tide gauge data. Section 3.2 has been rewritten and extended.

P9 124: "the superiority of BP measurements" : this point is not clearly demonstrated here. Need a spectrum of TG as in figure 3 + see next point.

The point is demonstrated clearly by the 'five times larger' in the previous comment and from inspection of the error bars in Figures 5 and 8. However, we agree that it is important to include additional spectra for the TG data, we should have done that before, please see our answer to the next point and to Reviewer 2 point 4.

P9 125-27: clearly modelling the non tidal variability should improve the results, you should make the test. You can use the Dynamic Atmospheric Correction (which is a barotropic modelling) to check this impact (the data are available on line on the AVISO website) or use NEMO as in page 12.

Many thanks for this comment which is similar to that of Reviewer 2 point 4. We have modified and expanded the discussion of Vernadsky data using DAC corrections in Section 3.2.

P10111: add references for FES2014

The web site (dated 2018) from which FES2014 can be downloaded is referenced on page 10 as FES2014 (2018) and given in the reference list. This is admittedly a strange way of referring to something, with a date in the model name which is not the date it was obtained. If there is a better way of referencing the model we would be happy to use it.

P11 115: "typically 1-year long records" : for BP different lengths have been used isn't it ?

Not really. On page 4 (section 2) we explained that almost all the BPR records are from annual deployments (apart from the MYRTLE records). These redeployments happened at almost the same time of year, constrained by the schedules of the BAS ships, resulting in roughly 1-year records. The exact dates for each one can be found from the PSMSL web site.

P11 l25-26: comparison is ambiguous: did you choose the 185\_ contour because this is the closest to the observed average phase lag ? or do you really take the geometrically mid-passage contour ? need to clarify

We don't see how the comparison is ambiguous. These lines of text are just mentioning a general comparison of the average phase lag for Mm reported in Section 3.1, and given that there were a similar number of deployments north and south, compared to the average phase lag at mid-Passage from the FES2014 model anyone would conclude by inspecting

Supplementary Figure 2(d). We have added '(Section 3.1)' following the mention of  $177^{\circ}$ , which makes it a little clearer where the numbers come from. We have also added '(Section 3.1)' where Mf average phase lag is mentioned on page 10, and for Mt following the reviewer's comment on page 12 line 26 below.

P11 129-30: indeed for 92-99, Mf amplitudes are smaller for south deployments : : : is this N-S difference small enough to be not significant ?

There is a misunderstanding here. Page 11, lines 29-30 are discussing Mm and not Mf. They make the point that the corresponding Mf amplitudes are not so different to later ones. We suspect that the earlier and lower Mm values referred to are to do with being further east at the south end of the F-S line. This would be another artefact of our having to deal with a data set that has both spatial and temporal dependence.

P12 17: "use of 5day values of BP": is it a running 5 days average ? why not using 1-day as what is done on BP measurements ?

5-day values are a standard NEMO product, see Hughes et al. (2018). Anyway, here we are discussing Mm for which 5-day sampling should be just about adequate.

P12 110-11: ": : : correlations were weaker in the north : : :" : can you explain more ?

Not really. The weaker correlation with NEMO in the north is almost certainly to do with the higher eddy variability as demonstrated by Sheen et al. (2014). Although NEMO has eddies in it, it is unreasonable to expect a model such as that to explain all the details of that variability. It does a better job in the south. The different character of variability north and south is demonstrated in Figure 3 of Hughes et al. (2018), see also Supplementary Figure 3 in the present paper. This is already explained in the text.

P12 l24: why do you use different names for Mt/Mtm ?

We don't. Mtm is mentioned only once, to point out that the constituent has that name in the FES2014 model. Otherwise we call it Mt throughout. You will often find tidal constituents having different names in the literature when they have been studied by different people through the years.

P12126: same comment as for Mm, see above.

We have added '(Section 3.1)' after the mention of  $197^{\circ}$  to make it clearer.

P12 130: "similar to that obtained above for figure 7a": the estimation for figure 7a are not shown in the text above ? : : : to add

We see the problem here with the word 'above'. We meant a couple of pages above in section 3.1. The wording has been made clearer.

P12 132: idem for estimations on figure 7b

Ditto

P1313: likely true for old versions of tidal packages : : :

It is not a question of old versions of packages as such. All of them use different sets of constituents depending on the lengths of records analysed etc. The fact of the matter is that the mid-latitude heritage of much tidal research (e.g. Darwin/Doodson/Cartwright) meant that Mt was not normally included in the standard sets, although there is no real reason why it could not have been.

P13 128: "should be separable from Mf : : : given a year of data": have you performed some tests ? using a long time series and then a one year time series to be able to say that ?

There are no tests necessary. We have mentioned the periods of the additional constituents MSf, MSm and MSt and those of our main three Mf, Mm and Mt, and you can check that the pairs are all separable within a year by the Rayleigh criterion. Now, of course, measurement errors can make this procedure more uncertain. So we have added 'In principle' to this sentence.

P13 129 : you mean removing these small conxtituents using an ocean model and then analyzing the studies frequencies ? but ocean models might not be enough accurate for such small constituents : : : please clarify.

We have changed the text to read 'ocean tide models' instead of 'ocean models' which may have been misleading. Inference of smaller constituents is a standard procedure in tidal analyses. The details can be left to whoever in the future does the analysis.

P14 13: + this point might also explain the different behaviours of BP and TG ?

We are not sure about this. We never combine BP and TG data in any fit, and within the TG fit alone the same 'k' factor will apply.

P14 18: "our determination of Mm": why not other components Mf, Mt ? please explain

Because Mm is the lowest of the three in frequency and the spectrum is red (or pink). Then please see the previous sentences.

P15 1 17: "stacks of records" : please explain

'Stacks' are when many records are combined into an overall fit. It is a technique often used in seismology, for example, and was used by Trupin and Wahr (1990) to look at long-period aspects of tide gauge records. Anyone interested can read that paper. We are not digressing into a discussion of that here.

P16 A2: you get these formulae from eq 2 and A1 ?

Yes. That is correct.

P17 111: how do you choose R=0.414 ?

We did not choose this value ourselves. It is the value of the amplitude of the main sideband in the tidal potential. However, we agree this was unclear and have added the Cartwright and Tayler (1971) and Cartwright and Edden (1973) references again.

P18 15: It is not clear why you choose to use simplified formulae in this paper ? explain please.

Because they are quite adequate for the simple nodal variations we are looking at here, especially given the uncertainties in the data. These simple forms are also the ones that normally appear in text books on tides. But there are applications such as Ray and Egbert (2012) where you have to take more complete ones. They are never completely correct, however, as they are trying to provide simple algebraic expressions of combinations of multiple sidebands in the tidal potential.

P18 113: R=0.065 ?

Yes. Again, this is a value that comes from the amplitudes for the sidebands of Mm in the tidal potential. The Cartwright tables (Cartwright and Tayler, 1971) were already referred to but we have added Cartwright and Edden (1973) to be more complete. We hope this is clearer now.

Legend of Figure 5:"one standard error" : please give a bit more details.

The caption has been expanded.

Technical corrections:

P1 1 16: replace by "while the phase difference for Mm"

Done. Many thanks.

P2 127: replace by "seems to be a good theory"

Done

P4 118: replace has -> have

Not done. Bottom pressure (BP) is singular.

P13 l22: replace will -> may

In our opinion, 'will' is better as these other constituents are bound to be present to some extent and there is no 'may' or 'perhaps' about it.

Reviewer 2

We are grateful for the time that both reviewers spent on this paper. The comments of Reviewer 2 are given below followed by our replies. The page and line numbers refer to the version in OS Discussions.

1. P1 31: 't=0,' =>'t=0. Eq[1] is further modified to;'?

Thanks but we think it reads ok as it is.

2. P6 6-8: Are the daily mean values suitable for resolving Mt (9.13 days)? Using daily means might be one of reasons for large error bars in Fig7. Though there are some discussions on the complications of this analysis approach, it will be better if authors can further discuss(/investigate) the dependence of long-period tides, especially Mt, on data temporal resolution.

We understand this comment but daily means (Nyquist period of 2 days) should be adequate for study of a cycle with a period of 9 days, irrespective of the small signals and large error bars in Figure 7. It was anyway convenient for us to use daily means which were a product of the Weighing the Ocean project. We don't feel a discussion suggested by the Reviewer's last sentence is warranted.

3. P7 4-7, Fig4 and other places in text: why is non-tidal variability larger in the south side as however there are more eddies in the north as authors mentioned in P12 5? MJO (intraseasonal) is taken as one potential contributor but it seems to me that there are still some significant features at longer timescales (e.g. in Fig4b between 350day and 450day, between 510day and 610 day). A bit more explanations/speculations are suggested to add here.

The Reviewer has misunderstood our purpose in showing Figure 4(a,b). Mf is the largest of the long-period tides discussed here and it varies a lot over the nodal cycle, hence the ratio of tidal to non-tidal variability varies over the cycle. We wanted to include Figure 4(a,b) as examples of that variability, when the tidal component was large and small respectively. The two plots (a,b) were not intended to show north/south differences in variability as such. We have added some words to make that clear.

As we mentioned, there is a lot of non-tidal variability due to eddies etc. in the north (Sheen et al., 2014) but also some in south. This results in spectra parameterised as shown in the Supplementary Material Figure 3 and discussed in Hughes et al. (2018). The MJO was mentioned in particular because the timescale of its variability is not too different from the Mm period. The features pointed out in Figure 4b are indeed interesting – they are presumably associated with rapid (non-tidal) ocean variability of some kind, there are odd features like that in many BP records that we have not investigated in detail. Fortunately, the stationary signals of the tide are fairly immune to such things.

4. Subsection 3.2: It compares the long-period tides derived from BPR and also the tide gauge record. How the power spectral distributions differ between those two kinds of records? If BPR has advantages of resolving long-period tides over tide gauge data, due to less non-tidal variability, one may expect there are more noises close to the 3 constituents' frequencies (Fig3). Is this true? It's good to show this merit.

The Reviewer is right, and this point relates to two by Reviewer 1 (P9 124 and P9 125-27). We have added spectra for Vernadsky for comparison. We have also considerably extended the discussion of Vernadsky data in Section 3.2 by using DAC corrections, and have added extra words to the Conclusions. Many thanks for suggesting we do this, which we should have done before.

5. P8 1-3 and Fig5b: It's worth proposing some explanations why such north-south differences are observed here, when this is not expected from the theory.

At this point in the paper, the different phase lags north and south demonstrate spatial variation unexpected from the Equilibrium tide, that (if correct) would have to be explained

by dynamical tidal differences. That is what we investigate further in the discussion of Section 4 by making use of the FES2014 model which, if you read on, are explained well by the model.

6. Fig 6&7: small amplitudes and large error bars make it difficult to detect the nodal cycle. It seems to me that error bars are slightly larger in the 1st decade. Is this related to BPR data density used? It'll be good if the data availability (after QC) of such 45 BPR records is provided.

It is true that error bars are slightly larger in the first decade, but it depends which plot you mean. For example, for the amplitude of Mt (Figure 7a) they are much the same, but the amplitudes themselves are larger in the second part which results in the errors on the phases (Figure 7b) being smaller. See also our reply to Reviewer 1 (P8 133). But in general this comment is right , we have no simple explanation, presumably it depends on the mix of locations (e.g. the F-S line in the early years), ocean variability changes in time etc.

Most of the BPR records were re-QC'd as part of the Weighing the Ocean project and are available on the PSMSL web site. All of them were inspected for possible glitches in the time series. We have put some wording in the Acknowledgements for anyone who would like copies of the data.

7. Author used FES2014 model to discuss the spatial variation of long-period tide parameters. FES models are assimilated by satellite altimeter data, which to me however have some limitations at high latitudes. Is this a noticeable concern here?

We don't think so. The orbit of TOPEX and its follow-on missions was designed to include the Drake Passage, and there is now over 25 years of precise altimetry, so there has been plenty of data for assimilation. Anyway much of the non-equilibrium dynamics of the longperiod tides can be modelled from first principles without assimilation (some references are given in the paper). We suspect that FES2014 is a very good model for the long-period tides. Anyway, it is certainly the best available for use here.

**The nodal dependence of long-period ocean tides in the Drake Passage**

Philip L. Woodworth1, Angela Hibbert1

[revised manuscript text omitted]

Vernadsky tide gauge data have been used in several studies of ACC variability alongside the information from the Drake Passage BPRs (Hughes et al., 2003; Woodworth et al., 2006). For present purposes, Vernadsky data enable an interesting comparison to be made on how much better Mf can be observed in BP measurements than in coastal tide gauge data. It might be supposed that Vernadsky data would have an advantage in being all from the same location, rather than at different positions for the BPR deployments. On the other hand, a coastal tide gauge record will clearly contain a considerable amount of non-tidal variability due to storm surges etc.

Each year of hourly data from Vernadsky has been Figure 3(c) shows the spectrum of sea level variability at Vernadsky. Comparison with Figure 3(a) demonstrates an order of magnitude larger amount of non-tidal background in Figure 3(c), with only Mf observed clearly, only a hint of Mm, and Mt hidden within the background. Each year of hourly data from Vernadsky was analysed in a similar way as described for the BP measurements, providing daily values of sea level from which estimates of Mf amplitude and phase lag have beenwere obtained. (We(Given the high noise levels at Mm and Mt frequencies in Figure 3(c), we considered Mm and Mt similar analyses for them to be below noise level in these oneyear records unfeasible.) Figure 8(a) shows the amplitude values, which have individual uncertainties approximately five times larger than for the BPRs in Figure 5(a). The mean amplitude in the plot is  $2.90 \pm 0.25$  cm (and so the Mf harmonic constant would have an amplitude of 2.90/1.043 = 2.78 cm). This is larger than for the nearby BPRs. The nodal cycle shown in red has an amplitude of  $1.20 \pm 0.36$  cm, or  $41 \pm 12$  % of the mean value, almost exactly the same as for the BPRs and again consistent with expectations from Equation 4. Phase lag (Figure 8(b)) is also consistent with the BP data, in having an average of  $184.9 \pm$ 4.7°. Within the large scatter from year to year, a nodal variation with an amplitude of  $22.1 \pm$  $7.5^{\circ}$  can be just about discerned. (Five years of data with phase lags outside the plot limits were not used in this nodal fit.)

ComparisonsTherefore, comparisons of Figures 5 and 8 underline the point we wish to make regardingdemonstrate the superiority of BP measurements to compared to coastal tide gauge records in long-period tidal studies. It is possible that modelling of, unless the nontidal variability in the Vernadsky records in terms of a response to winds and air pressures could result in a more clearly identified Mf, but one doubts if it could everbackground in the latter can be equally as good.modelled efficiently. Crawford (1982) provides an earlier example of an attempt at such modelling in Canadian tide gauge data.

Fortunately, Dynamic Atmospheric Correction (DAC) data sets are now available which provide estimates of the sea level response to air pressures and winds every 6 hours on a 0.25° global grid. Estimates are based on the use of a high-resolution barotopic model for high-frequency variability (timescales less than 20 days) and the assumption of the inverse barometer response for longer timescales. Details are available from the Archivage, Validation et Interprétation des données des Satellites Océanographiques (AVISO) web site (https://www.aviso.altimetry.fr). Carrère and Lyard (2003) demonstrated how effective such modelling could be in estimating non-tidal variability in tide gauge records.

Figure 3(d) shows the spectrum of sea level variability at Vernadsky once the DAC correction has been applied. Complete years of DAC corrections are available for 1993-onwards. Therefore, they have been employed for the 22 years 1993-2014 only. Comparison

to Figure 3(c) shows that most of the background has been modelled effectively, down to a level little greater than that for the BPRs in Figure 3(a), and that Mf, Mm and Mt can now all be clearly identified above the background. Figure 8(c-h) contains a set of analyses of nodal variations for Mf amplitude and phase lag (c,d), Mm (e,f) and Mt (g,h), all based on the DAC-corrected data for 1993-2014. In the case of Mf (Figures 8c,d), the mean amplitude is  $2.59 \pm 0.13$  cm (and so the Mf harmonic amplitude is 2.59/1.043 = 2.49 cm). The nodal cycle red has an amplitude of  $1.05 \pm 0.19$  cm, or  $41 \pm 7\%$ , about the same as for the uncorrected data in Figure 8(a). Average phase lag (Figure 8d) has a value of  $207.7 \pm 2.9^{\circ}$  (approximately  $23^{\circ}$  larger than for the uncorrected data), and now a clear nodal cycle can be seen with an amplitude of  $23.4 \pm 4.0^{\circ}$  (without the need to reject any values for being outside plot limits).

In the case of Mm (Figure 8e), the average amplitude is  $1.47 \pm 0.13$  cm, while the nodal fit has an amplitude of  $10 \pm 13$  % of the average but with the opposite sign expected from Equation 8. This finding is similar to the difficulty of explaining Mm amplitude from the BPR data in Figure 6(a) reported above, and discussed further in the following section. Phase lag for Mm (Figure 8f) has an average value of  $174.0 \pm 6.8^{\circ}$ , with no evident nodal variation as expected from Equation 8. The average amplitude of Mt (Figure 8g) is  $0.57 \pm 0.13$  cm with a nodal variation of  $13 \pm 34\%$  of the mean, while Figure 8h shows an average phase lag of  $232.6 \pm 12.4^{\circ}$  and a nodal amplitude of  $47.3 \pm 19.2^{\circ}$ .

Overall, one can see the benefit of using the DAC corrections. The non-tidal variability in the sea level spectrum is much reduced, and nodal variations in all three long-period tides can now be investigated more reliably. Mf and Mt amplitudes and phase lags, and Mm phase lag, are generally consistent with equilibrium expectations, Mm amplitude being an exception to be discussed further below. All the above Vernadsky findings are summarised in Table 1.

**4** Discussion**

Some of the findings of the previous section are consistent with expectations from the equilibrium tide, while those that are not require explanation.

As mentioned above, the long-period tides in the equilibrium tide have simple spatial distributions in amplitude and phase, with north-south variations only. However, their spatial distributions in the real ocean are now known to depart considerably from equilibrium expectations, with larger departures at shorter period (e.g. see Figure 2 of Ray and Erofeeva, 2014). These differences are most evident when contrasting the Pacific, Atlantic and Indian Ocean low- and mid-latitude basins.

If one considers Mf in particular, atlases of this constituent have been available for many years, notably since the data assimilation numerical modelling of Schwiderski (1982). More recent co-tidal distributions for Mf have been obtained from altimeter measurements and models by Kantha et al. (1998, Figure 7), Mathers and Woodworth (2001, Plate 4) and Egbert and Ray (2003, Figure 1). These are consistent with Mf phase lag increasing when travelling south down the Pacific coast of South America, with the 180° contour around the Drake Passage, and with a complicated amphidromic pattern in the South Atlantic to the NE of the Falklands. More recent studies have included the development of the FES2004 ocean tide model, which also showed these features (Lyard et al., 2006, Figure 2), with roughly the same Mf amplitude on both sides of the Drake Passage and larger phase lag on the south side than north side.

FES2014 (Finite Element Solution 2014) is the latest in the series of state-of-the-art global ocean tide models provided by French groups. It provides elevations and currents (amplitude and phase) and tidal loading information for 34 tidal constituents on a global 1/16°x1/16° grid. FES2014 (2018) provides more detailed information.

Supplementary Figure 2(a,b) shows the Mf amplitude and phase lag for Mf at the Drake Passage from the FES2014 model. Some points of consistency with our findings are as

follows. First, the model has much the same amplitude over the whole area (~2 cm), and phase lags are essentially zonal, largely justifying our decision to combine amplitudes and phase lags from all deployments in Figure 5, and the subsequent discussion in terms of north-and south-side values.

Second, we found the amplitudes for Mf to be similar on the north and south sides of Drake Passage (Figure 5a), but phase lags were shown to be  $22 \pm 2^{\circ}$  larger for the southern deployments (Figure 5b). The latter is qualitatively consistent with Supplementary Figure 2(b). Third, the 192° average phase lag for Mf from all the BPRs taken together (Section 3.1, Figure 5b) is consistent with the ~190° contour in mid-Passage in Supplementary Figure 2(b). OnIn addition, the other hand, the 185° phase lag atMf harmonic constants estimated above for Vernadsky is a little lower than the ~200° one would infer from Supplementary Figure using DAC-corrected data (2(b.49 cm amplitude and 208° phase lag) are similar to those in FES2014 (2.41 cm and 202° respectively). (FES2014 amplitudes and phase lags for Mm and Mt (1.31 cm and 190° and 0.42 cm and 211° respectively) are all consistent with DAC-corrected Vernadsky findings to within ~1 or ~2 standard deviations for amplitudes and phase lags respectively).

[revised manuscript text omitted]
. Section 3.2 showed that, when Vernadsky coastal tide gauge data were corrected for non-tidal variability, then a major improvement in identification of the long-period tides results (e.g. reduction in the uncertainties for Mf in Table 1 by a factor of two). However, the Drake Passage BPRs, which were located in deeper water where the inverse barometer-related sea level variations are compensated automatically by BP itself, have still provided more accurate estimates of nodal variation, in spite of different locations for deployments. Table 1 demonstrates that the uncertainties for Vernadsky Mf, even when DAC-corrected, are still double those of the FES2014-corrected BPRs. (A similar conclusion can be obtained from inspection of the uncertainties displayed in Figures 5(c,d) and 8(c,d)). Nevertheless, it is the case that there is a lot more tide gauge data available for study worldwide than BPR data (Woodworth et al., 2017). Therefore, an obvious recommendation following from the present work is that tide gauge data be investigated more completely in order to investigate whether the temporal variation of long-period tides conforms to equilibrium expectations, perhaps by employing 'stacks' of records, as has been used previously to investigate other long-period components of tide gauge records (e.g. Trupin and Wahr, 1990).), with DAC-type corrections applied to each record.

**Competing interests**

The authors declare that they have no conflict of interest.

Acknowledgements. The programme of Drake Passage bottom pressure measurements was led by scientists from the National Oceanography Centre including Ian Vassie, Mike Meredith, Chris Hughes and Miguel Ángel Morales Maqueda. The bottom pressure recorders were designed, constructed and deployed by Bob Spencer, Peter Foden, Jeff Pugh, Steve Mack, Geoff Hargreaves and others. The help of the British Antarctic Survey with the deployments is much appreciated. Data sets were processed by David Blackman-and, Philip Axe and Angela Hibbert, and may be obtained from the web sites mentioned in Section 2 or from the authors of this paper. 
[revised manuscript text omitted]

- Carrère, L., and Lyard, F.: Modeling the barotropic response of the global ocean to atmospheric wind and pressure forcing - comparisons with observations, Geophys. Res. Lett., 30, 1275, doi:10.1029/2002GL016473, 2003.
- Cartwright, D.E., and Tayler, R.J.: New computations of the tide-generating potential, Geophys. J. Roy. Astr. S., 23, 45-74, doi:10.1111/j.1365-246X.1971.tb01803.x, 1971.
- Cartwright, D.E., and Edden, A.C.: Corrected tables of tidal harmonics, Geophys. J. Roy. Astr. S., 33, 253-264, doi:10.1111/j.1365-246X.1973.tb03420.x, 1973.
- Cartwright, D.E., Spencer, R., and Vassie, J.M.: Pressure variations on the Atlantic equator, J. Geophys. Res., 92(C1), 725-741, doi:10.1029/JC092iC01p00725, 1987.
- Cartwright, D.E., Spencer, R., Vassie, J.M., and Woodworth, P.L.: The tides of the Atlantic Ocean, 60N to 30S, Philos. T. R. Soc. Lond., A, 324, 513-563, doi:10.1098/rsta.1988.0037, 1988.
- Cartwright, D.E.: Tides: a scientific history. Cambridge University Press: Cambridge, 292pp, 1999.
- Crawford, W.R.: Analysis of fortnightly and monthly tides, Int. Hydrogr. Rev., LIX, 131-141, 1982.
- Doodson, A.T.: The harmonic development of the tide-generating potential, Philos. T. R. Soc. Lond., A, 100, 305-329, doi:10.1098/rspa.1921.0088, 1921.
- Doodson, A.T.: VI. The analysis of tidal observations, Philos. T. R. Soc. Lond., A, 227, 223-279, doi:10.1098/rsta.1928.0006, 1928.
- Doodson, A.T., and Warburg, H.D.: Admiralty Manual of Tides, His Majesty's Stationery Office, 270pp, 1941.
- Egbert, G.D., and Ray, R.D.: Deviation of long-period tides from equilibrium: kinematics and geostrophy, J. Phys. Oceanogr., 33, 822-839, doi:10.1175/1520-0485(2003)33<822:DOLTFE>2.0.CO;2, 2003.
- Feng, X., Tsimplis, M.N., and Woodworth, P.L.: Nodal variations and long-term changes in the main tides on the coasts of China, J. Geophys. Res. Oceans, 120, doi:10.1002/2014JC010312, 2015.
- FES2014: Description of the FES2014 ocean tide model, obtained from https://www.aviso.altimetry.fr/en/data/products/auxiliary-products/global-tide-fes/description-fes2014.html. Downloaded January 2018.
- Foreman, M.G.G., and Neufeld, E.T.: Harmonic tidal analyses of long series, Int. Hydrogr. Rev., LXVIII, 85-108, 1991.
- Gross, R.S.: An improved empirical model for the effect of long-period ocean tides on polar motion, J. Geod., 83, 635–644, doi:10.1007/s00190-008-0277-y, 2009.
- Hibbert, A., Leach, H., Woodworth, P.L., Hughes, C.W., and Roussenov, V.M.: Quasibiennial modulation of the Southern Ocean coherent mode, Q. J. R. Meteor. Soc., 136, 755-768, doi:10.1002/qj.581, 2010.
- Hughes, C.W., Woodworth, P.L., Meredith, M.P., Stepanov, V., Whitworth, T., and Pyne A.R.: Coherence of Antarctic sea levels, Southern Hemisphere Annular Mode, and

[revised manuscript text omitted]